# Maintenance of appropriate size scaling of the *C. elegans* pharynx by YAP-1

Klement Stojanovski[1], Ioana Gheorghe[1,2], Peter Lenart [1], Anne Lanjuin[3], William B. Mair [3] & Benjamin D. Towbin [1] ✉

Even slight imbalance between the growth rate of different organs can accumulate to a large deviation from their appropriate size during development. Here, we use live imaging of the pharynx of *C. elegans* to ask if and how organ size scaling nevertheless remains uniform among individuals. Growth trajectories of hundreds of individuals reveal that pharynxes grow by a near constant volume per larval stage that is independent of their initial size, such that undersized pharynxes catch-up in size during development. Tissue-specific depletion of RAGA-1, an activator of mTOR and growth, shows that maintaining correct pharynx-to-body size proportions involves a bi-directional coupling between pharynx size and body growth. In simulations, this coupling cannot be explained by limitation of food uptake alone, and genetic experiments reveal an involvement of the mechanotransducing transcriptional co-regulator *yap-1*. Our data suggests that mechanotransduction coordinates pharynx growth with other tissues, ensuring body plan uniformity among individuals.

The correct size and proportions of organs are crucial for organismal function, and disproportionate organs are associated with a large diversity of diseases[1–4]. However, attaining correct size proportions is challenged by fluctuations in organ growth rates. During development, organs grow by orders of magnitude, such that even small deviations from the correct growth rate could in principle amplify to large deviations in size over time[5,6]. While such size divergence of organs usually does not occur, the mechanisms that prevent size divergence are not understood.

At the scale of individual cells, size homeostasis has been extensively studied by time-lapse microscopy of yeasts, bacteria, and mammalian cells. In cells, stochastic size fluctuations are corrected within a few cell divisions. For many cell types, larger cells on average undergo a smaller volume fold change per cell cycle than smaller cells. Thereby, cells that deviate from the norm return to a stable reference point. Depending on how fast this reference point is reached, cells are called to follow adder or sizer mechanisms[7–11]. A sizer refers to cell types that, on average, return to the appropriate size within one cell cycle such that their size at division is independent of their size at birth.

An adder refers to cells that, on average, grow by a constant absolute volume, independent of their size at birth. Unlike sizers, adders take multiple cell cycles to return to a reference point.

At the scale of an entire organism, our recent work using *C. elegans* showed that size uniformity among individuals is maintained by a mechanism distinct from sizers and adders[12]. The fold change in the body volume of *C. elegans* per larval stage is nearly independent of the size at the beginning of a larval stage. Nevertheless, rapidly and slowly growing individuals diverge only weakly in volume during development due to an inverse coupling of their growth rate to the duration of their development. This coupling is mediated by a genetic oscillator through a mechanism we termed a growth-coupled folder[12].

At the scale of organs, size proportions follow robust allometric relations[13,14], and proportions of body parts usually scale appropriately over a wide range of body sizes[15]. Combined experimental and theoretical work provides elegant explanations for how tissue patterning appropriately scales with tissue size during development[16,17]. For example, a negative feedback between two diffusible components can ensure scale invariance of morphogen gradients[15–18]. Genetic

---

[1]Institute of Cell Biology, University of Bern, Bern, Switzerland. [2]Graduate School for Cellular and Biomedical Sciences, University of Bern, Bern, Switzerland. [3]Department Molecular Metabolism, Harvard TH Chan School of Public Health, Boston, MA, USA. ✉e-mail: benjamin.towbin@unibe.ch

experiments revealed tissue autonomous and systemic mechanisms that control organ size[19]. For instance, morphogen gradients are thought to limit the lateral expansion of imaginal discs in *Drosophila melanogaster*[20,21] and damaged imaginal discs trigger systemic responses via secretion of the relaxin-like signalling peptide *Dilp8*[22,23]. Similarly, unilateral inhibition of limb growth in mice triggers a growth response that retains proper limb symmetry[24,25]. However, individual organ growth trajectories have rarely been measured in vivo over time, and how deviations in organ size are dynamically corrected during development remains poorly understood.

The YAP/Hippo signaling pathway[26,27], which transduces mechanical stimuli to cellular responses[26] plays an important role in organ growth control. Hippo kinase is a negative regulator of the transcriptional regulators YAP and TAZ (jointly referred to as YAP from here onwards), which activate genes involved in cell proliferation and cytoskeleton organization[28]. Tissue-specific mutation of Hippo kinase in fly imaginal discs and the mouse liver leads to ectopic activation of YAP and to oversized organs[29–35]. However, loss of YAP does not substantially impact organ growth or size, suggesting that it is not an immediate activator of cellular growth[36]. How YAP contributes to the instruction of organ size during normal development therefore remains an important open question.

In summary, despite the identification of molecular factors that impact organ size, the mechanisms by which the many tissues of an organism coordinate their growth to robustly achieve the correct size proportions are far from understood. Classic genetic approaches to this question often face the challenge of pleiotropic phenotypes that are difficult to dissect mechanistically. To overcome this challenge, we devised live imaging methodology using *C. elegans* to precisely quantify fluctuations of growth close to the endogenous state and minimize the impact of pleiotropic effects. We use the pharynx as a model and show that this organ converges to a size setpoint by an adder-like mechanism, reminiscent of size control mechanisms previously observed for cells[7–11]. However, unlike cellular adders, pharyngeal size control involves the systemic and bi-directional coupling of pharynx growth to other tissues via the mechano-transducer YAP/*yap-1*. Importantly, knock-down of *yap-1* did not impair the speed of pharynx and body growth under otherwise unperturbed conditions, but *yap-1* was specifically required for the robustness of pharynx size proportions to an imbalance in tissue growth. We conclude that while *yap-1* is dispensable for rapid organismal growth per se, it plays a crucial role in the coordination of the growth of the pharynx and other organs.

## Results

### Quantification of pharynx and body growth of *C. elegans* by live imaging

To quantify the growth and size of the pharynx relative to other tissues, we created a *C. elegans* strain expressing a green fluorescent protein in the pharyngeal muscle (*myo-2p::gfp*), and ubiquitously expressing a red fluorescent protein (*eft-3p:mscarlet*) (Fig. 1a). We recorded growth of hundreds of individual animals of this strain at 25 °C in micro chambers using a temperature-controlled fluorescence microscope at a time resolution of 10 min[12]. By automated image analysis[12], we determined the length of the pharynx and the total body at each time point from planar optical sections (Supplemental Fig. 1a), and estimated their volumes based on their near rotational symmetry (Fig. 1b)[37–40]. Measurement noise for individual animals was lower than 1.9% for the pharynx and lower than 1.2% for the total body from the first molt (M1) onwards, and <5% between hatching and the molt (Supplementary Figs. 1c–f, 2).

As we previously observed[12], total body growth was halted during four periods of approximately two hours, corresponding to lethargic phases prior to cuticular moulting (Fig. 1b). We could thereby automatically detect larval stage transitions by the time points at which

growth and feeding resumed and compute growth rates ($\mu$) of body and pharynx at each larval stage as the increase in log(volume) per time. Throughout this article, growth rate refers to the change in log transformed volume per time ($\mu = d\log(\text{vol})/dt = (d\,\text{vol}/dt)/\text{vol}$), i.e., the growth rate normalized to the current size, unless specified as the absolute growth rate ($\mu_{abs} = d\,\text{vol}/dt$), which indicates the absolute change in volume per time.

The average growth rate of the pharynx was about half the growth rate of the total body volume (volume doubling times of 12.02 +/− 0.94 vs. 6.64 +/− 0.51 h). Consistently, the cumulative volume fold change from hatch to the fourth moult (M4) was six times larger for the total body (53.3 +/− 5.3 fold) than for the pharynx (9.02 +/− 1.05 fold). Similar to allometric growth of the head and body of many animals, the volume fraction of the pharynx thus declined from 18 +/− 1.7% at hatching to 3.1 +/− 0.27% at M4.

### Pharynx size heterogeneity does not increase during development

During development, small differences in the growth rates among individuals can quickly amplify to significant differences in volume due to the compound effects of exponential growth. Our previous work showed that the coefficient of variation (CV) of the body volume among individuals nevertheless remains below 10%[12] (CV of body length: ~4%) (Fig. 1c, Supplementary Fig. 1c, e) due to an inverse coupling of growth to the duration of larval stages[12].

The CV of the pharyngeal volume (~4%) and length (~2%) was even smaller than that of the total body size (Fig. 1c, Supplementary Fig. 1c, e), except for volume measurements immediately after hatching (~8%). This larger observed CV at hatch is, at least partially, explained by the larger technical noise in volume and length measurements at this early stage (Supplementary Figs. 1c–f, 2). At other stages, the impact of technical noise on the observed CV of pharynx and body volumes was near negligible (Supplementary Figs. 1c, d, 2).

To determine whether the observed volume heterogeneity at the end of development was smaller than expected given the heterogeneity of the pharyngeal growth rates, we ran simulations in which we randomly shuffled the measured growth parameters among individuals. We then compared the experimentally measured volume heterogeneity to the heterogeneity produced by these randomized simulations. Simulations were started at M1 to exclude potentially confounding effects of non-negligible measurement noise at very early stages. Simulations starting at hatch yielded consistent results and are shown as supplemental information (Supplementary Fig. 1b).

Based on our previous work[12], we distinguished three scenarios: (1) independently randomizing the growth rates and larval stage durations among individuals (*uncoupled folder*), (2) randomly shuffling of the volume fold changes per larval stage among individuals, which retains the coupling between growth and larval stage duration (*coupled folder*), and (3) randomly shuffling the added volumes per larval stage among individuals (*adder*). Whereas simulations of folders produced substantially larger CVs than experimentally observed, the simulated adder produced a CV close to the observed heterogeneity (Fig. 1d). Together, these results show that the coupling of growth and development observed at the scale of the entire animal[12] is insufficient to explain the observed uniformity of pharynx volume, suggesting additional mechanisms that counteract the divergence of pharyngeal volume during development.

### Sub-exponential pharynx growth within one larval stage results in an adder-like behaviour

To test directly if the *C. elegans* pharynx follows an adder-like behavior, we analyzed the relation between the volume at the beginning of a larval stage ($V_1$) and the added volume within a given larval stage ($\Delta V$) for the pharynx and the total body. Under exponential growth, and in the absence of size-dependent control of growth, $\Delta V$ is expected to

correlate positively with $V_1$. A positive correlation was indeed observed for the total body volume of *C. elegans* (Fig. 2a, Supplementary Fig. 3a), except for L1 animals as we previously described[12]. However, for the pharynx, we did not observe a positive correlation between $\Delta V$ and $V_1$. Instead, the absolute volume increase $\Delta V$ during L2 and L4 stages was nearly independent of the starting volume $V_1$, reminiscent of adders observed in single-celled systems[7,9,10], whereas during L1 and L3, $\Delta V$ and $V_1$ of the pharynx were anti-correlated (Fig. 2b, Supplementary Fig. 3b). This apparent behavior of the pharynx close to an adder was not due to measurement noise and persisted after correcting for attenuation bias (Supplementary Fig. 3c). Consistently, simulations showed that the observed relation between $\Delta V$ and $V_1$ of the pharynx are incompatible with a folder model, even when considering measurement noise (Supplementary Fig. 3d, see methods for details). Thus, the weaker relation between $\Delta V$ and $V_1$ for the pharynx compared to the body is not due to technical, but due to biological differences. We, however, do not exclude a quantitative impact of measurement noise on the precise relation between $\Delta V$ and $V_1$, especially for L1 where the technical noise is largest.

To study the temporal dynamics of pharynx growth, we analysed the trajectory of the volume growth rate within each larval stage for

animals of different volume. We binned individuals according to their pharynx or body volume in the middle of each larval stage and calculated the absolute rate of volume increase ($\mu_{abs} = dV/dt$) as a function of larval stage progression. Consistent with autocatalytic and supra-linear body growth[12], $\mu_{abs}$ was correlated with the body volume and was larger at the end than at the beginning of each larval stage (Fig. 2c, Supplementary Fig. 4a). However, for the pharynx, $\mu_{abs}$ was nearly constant within a given larval stage and increased near step-wise between larval stages (Fig. 2c, Supplementary Fig. 4b). Moreover, large and small pharynxes of the same larval stage had nearly the same $\mu_{abs}$ (Fig. 2c, Supplementary Fig. 4a). We conclude that although pharyngeal volume growth appears near exponential when inspected across multiple larval stages (Fig. 1b, Supplementary Fig. 4b), the pharynx grows near linearly over time within each larval stage. Linear growth at an absolute growth rate $\mu_{abs}$ that is independent of the starting volume is expected to produce a size-independent volume increase $\Delta V$ per time ($V(t) = V_0 + \mu_{abs}*t = V_0 + \Delta V$), which is consistent with the observed adder-like behavior of the pharynx.

As described above, simulations of an adder can indeed reproduce the observed coefficient of variation in pharynx volume, whereas other models produce a volume divergence that is larger than what we

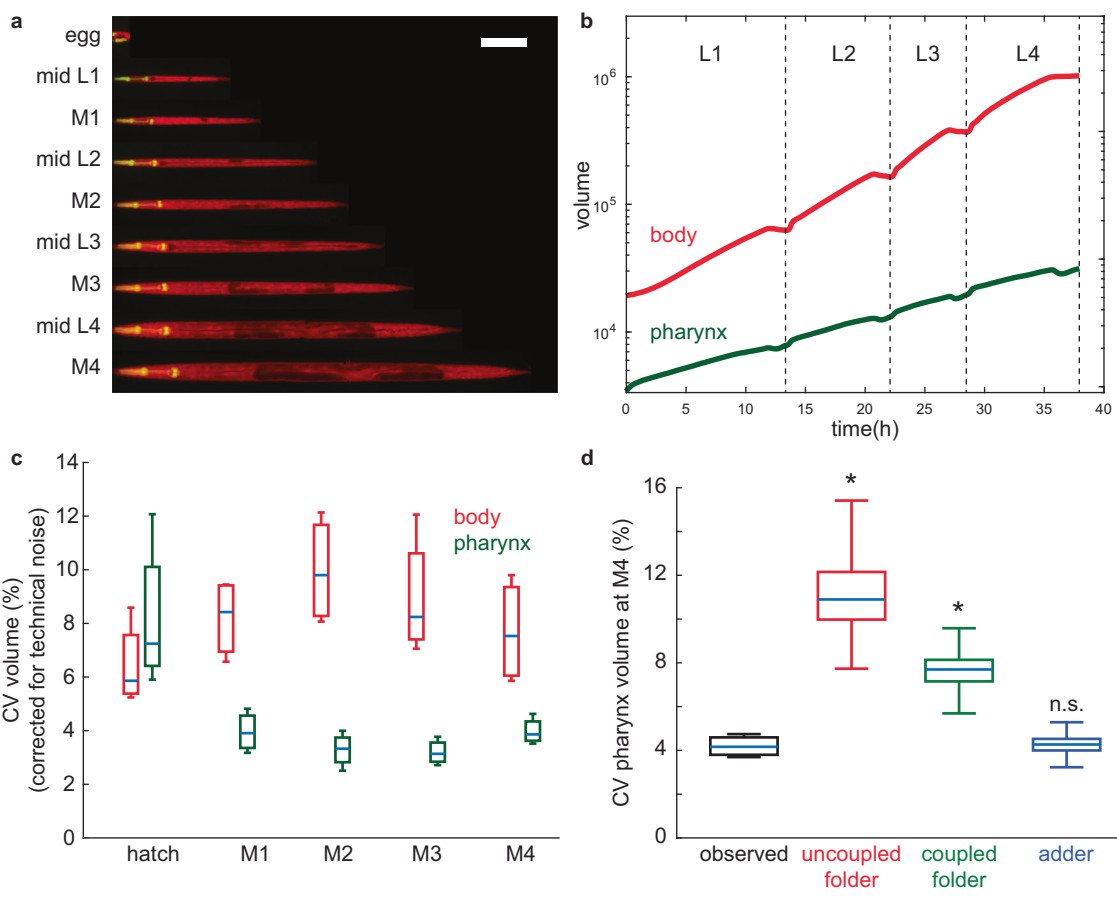

**Fig. 1 | Pharynx volume is less heterogeneous among individuals than total body volume. a** Individual animal imaged in micro chambers at indicated developmental milestones. Merged image of pharynx in green (*myo-2p:gfp*) and total body in red (*eft-3p:mscarlet*) is shown. Overlay of green and red is shown as yellow. *eft-3p:mscarlet* is comparatively weak in the head region in young L1 animals, such that the pharynx appears green at this stage. Contrast was adjusted for each time point individually. Animals were straightened computationally. Scale bar: 100 μm. **b** Body (red) and pharynx (green) volume as a function of time averaged from n = 475 individuals. For averaging, individuals of each larval stage were re-scaled to have matching larval stage entry and exit points and the growth curve was scaled back to the mean larval stage duration after averaging. **c** Coefficient of variation

(CV) of body (red) and pharynx (green) volume at hatch and larval moults (M1 to M4). Box plots represent CVs of m = 4 independent day-to-day repeats and n = 475, 641, 641, 638, 621 individuals for hatch, M1, M2, M3, M4. CVs were corrected for impact of technical noise. **d** Comparison of observed CV of pharyngeal volume to randomized simulations at M4. Simulations were initiated using volumes at M1 as described in the main text. p-value (ranksum test, two-sided) for observed vs. model: 0.03, except for adder: 0.49. boxplot show CV of m = 4 independent experimental populations. Number of individuals n as in (**c**). Boxplots in (**c, d**): central line: median, box: interquartile ranges (IQR), whisker: ranges except outliers (>1.5*IQR).

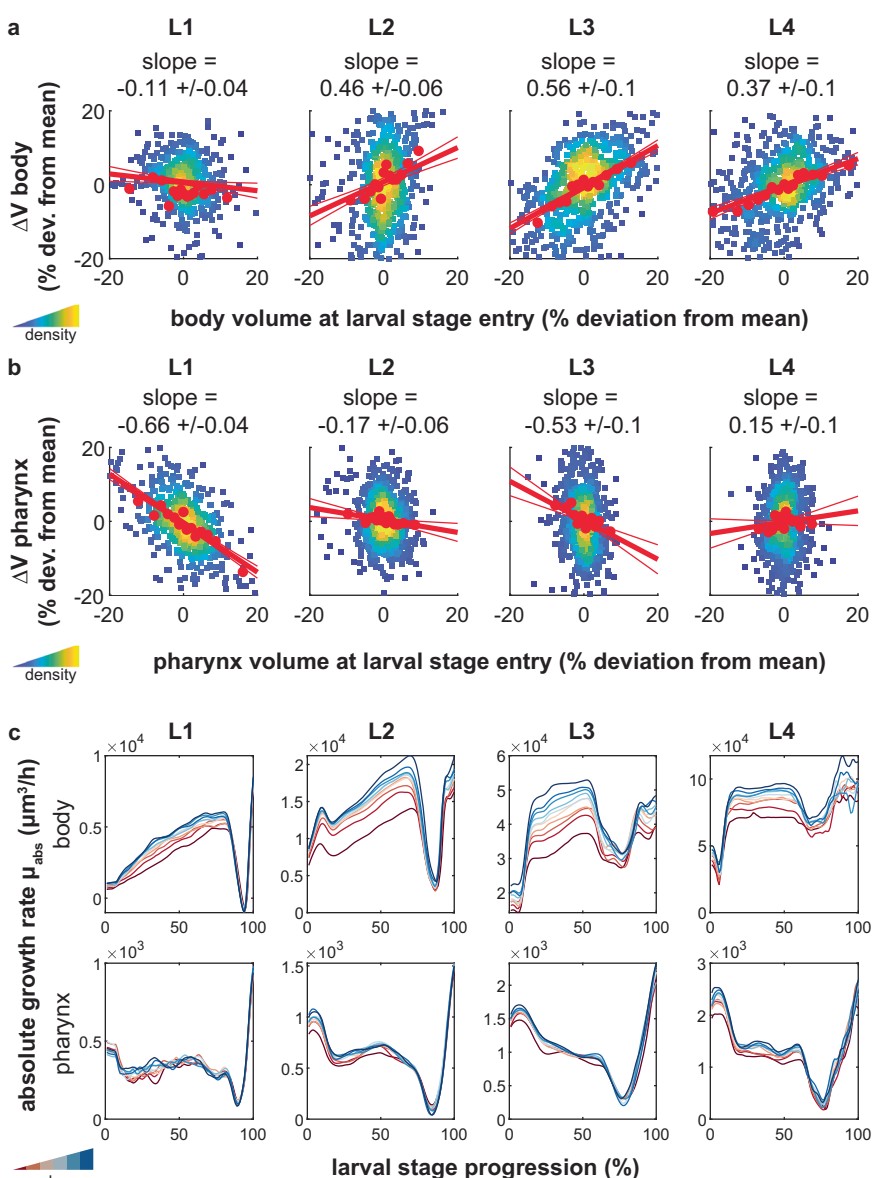

**Fig. 2 | Near linear pharyngeal volume growth within larval stages produces an adder-like behavior. a** Scatter plot of body volume at larval stage entry vs. added body volume per larval stage for each individual. Red circles: binned average along x-Axis. red line: robust linear regression (thick) with 95% confidence intervals (thin). Data is shown as relative deviation to the batch mean. Rare outliers beyond the +/− 20% range are omitted for clarity. Value above chart indicates slope of regression line +/− 95% CI. Color code indicates point density. **b** As (**a**), but for pharynx volume. **c** Absolute rate of body (top) and pharynx (bottom) volume growth as a function of larval stage progression. Individuals were binned into 10 classes according to their volume at 40% of each stage (red is the smallest volume, blue the largest volume). Individuals were re-scaled from moult to moult before averaging. The drop in growth rate at 80–90% of the larval stage represents the growth halt during lethargus. **a**–**c** $n$ = 475, 641, 641, 638 individuals for L1 to L4 from $m$ = 4 day-to-day repeats.

experimentally observed (Fig. 1d, Supplementary Fig. 1c). Notably, we do not observe a correlation between the pharynx volume and the body growth rate among individuals, suggesting that the pharynx volume is not limiting for growth in the observed range, and that the uniformity of pharynx volume is not a passive consequence of changes in the food uptake rate (Supplementary Fig. 4c).

**Auxin-inducible degradation of RagA/RAGA-1 allows quantitative titration of growth rates**

The adder-like behavior of the pharynx suggests that growth is controlled in a size-dependent manner to ensure pharynx size uniformity among individuals. However, observing an adder does not inform on the mechanism underlying this control. At least two distinct, but not mutually exclusive, classes of mechanisms are

conceivable. First, the narrow pharyngeal volume distribution could be due to precise, tissue-autonomous control. Second, pharyngeal volume scaling could involve crosstalk with other tissues. To distinguish between these two scenarios, we developed an experimental approach to perturb pharynx growth and other tissues by tissue-specific auxin-induced degradation (AID)[41] of the mTORC1 activator RagA/RAGA-1[42–44] (Fig. 3a).

We expressed the plant ubiquitin ligase Tir1 under the control of ubiquitous or tissue-specific promoters and modified the endogenous locus of *raga-1* with an *aid*-tag[44]. Supplementation of the plant hormone auxin (indole-3-acetic-acid, IAA) leads to dosage dependent ubiquitination of AID-tagged proteins and consequently their quantitative knock-down by proteasomal degradation in the tissue where Tir1 is expressed[45]. We could thereby perturb growth by depleting RAGA-1

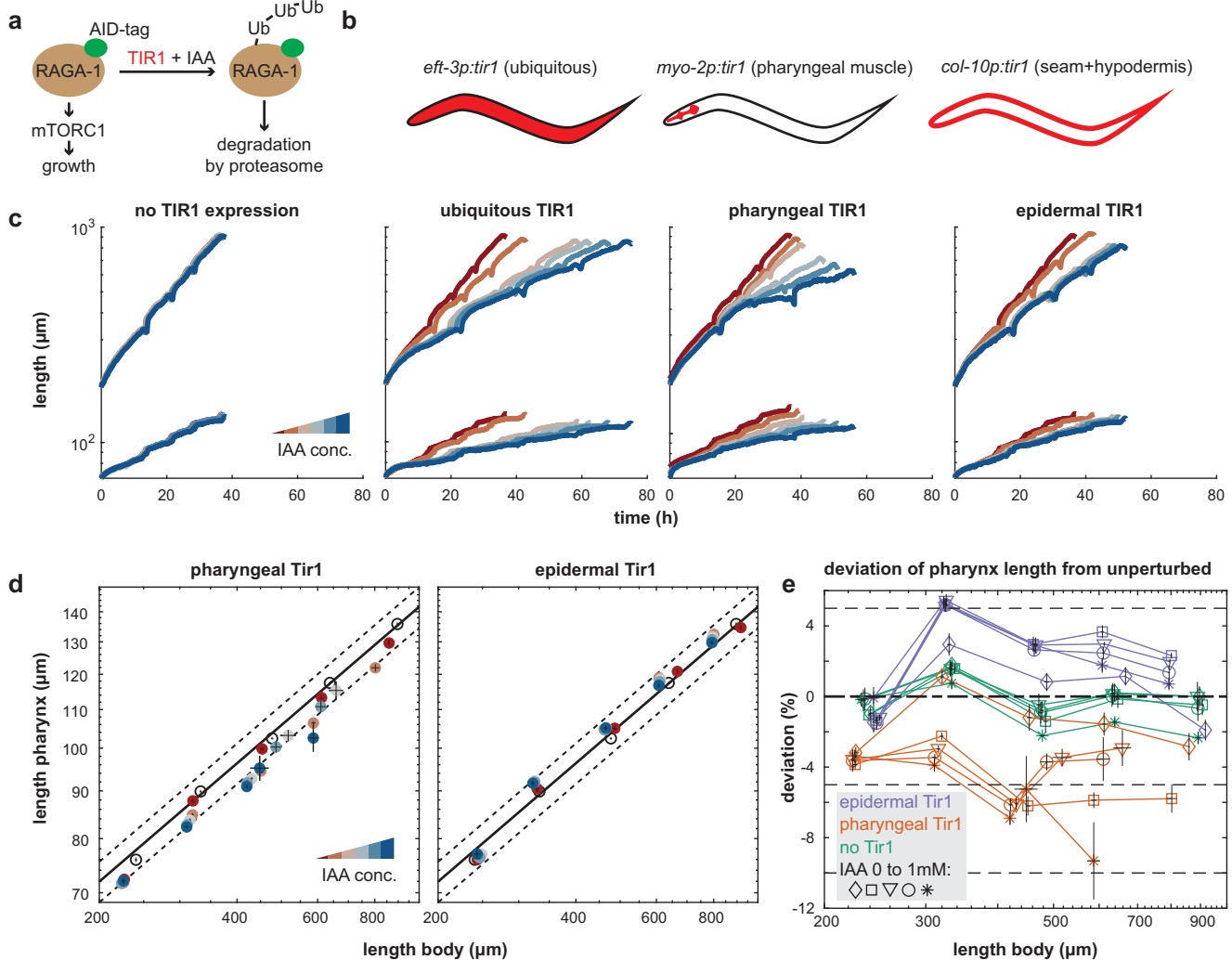

**Fig. 3 | Pharynx-to-body length proportions are robust to tissue-specific depletion of RAGA-1. a** Experimental approach to deplete RAGA-1 in selected tissues. An AID tag was inserted into the endogenous locus of the *raga-1* gene and Tir1 was expressed ubiquitously or under tissue-specific promoters. Addition of auxin to the growth medium leads to ubiquitination and proteasomal degradation of RAGA-1 in tissues expressing Tir1. **b** Schematic representation of tissue-specific promoters used to express Tir1. **c** Body (top) and pharynx (bottom) length as a function of time after depletion of RAGA-1 in indicated tissues. Color indicates IAA concentration from red to blue: no Tir1 expression 0 mM IAA, Tir1 expression + 0 mM, 0.1 mM, 0.25 mM, 0.5 mM, 1 mM IAA. **d** Scatter plot showing body vs. pharynx length at the beginning of L1 and at all larval moults M1 to M4 (circles in order from left to right) under pharyngeal (left) or epidermal (right) AID of RAGA-1. black circles: relation between pharynx and body length when unperturbed (no Tir1 expression and no IAA). Solid black line: linear regression to unperturbed body-to-pharynx length (P-line). Dashed black line: 5% deviation from P-line. Coloured

circles: IAA concentrations from red to blue: 0 mM, 0.1 mM, 0.25 mM, 0.5 mM, 1 mM. Error bars are standard error of the mean among day-to-day repeats. Where error bars are invisible, they are smaller than the marker. Slope *m* of P-line from 0 to 1 mM IAA for pharyngeal RAGA-1 AID: 0.43, 0.41, 0.43, 0.43, 0.37; for epidermal RAGA-1 AID: 0.43, 0.45, 0.45, 0.45, 0.43; without Tir1 expression: 0.44, 0.43, 0.43, 0.43, 0.42. **e** Deviation of pharynx-to-body length ratio (deviation from P-line) vs. body length at 30% of L1 stage and at larval moults (mean of day-to-day repeats). Colours indicate tissues of Tir1 expression; orange: pharynx, purple: epidermis, green: no Tir1 expression. Marker symbol indicates IAA concentration: diamond: 0 mM, square: 0.1 mM, triangle: 0.25 mM, circle: 0.5 mM, asterisk: 1 mM. Error bars are standard errors among day-to-day repeats, where invisible, errors are smaller than the marker. **c–e** Number individuals n, 26 < n < 259 from number of day-to-day repeats m between 2 and 10. See Supplementary Tables 1 and 2 for n and m of each condition.

by AID (Fig. 3a) and measure length and volume growth in micro chambers. Length and volume measurements led to equivalent conclusion. In the following, for their higher day-to-day reproducibility, length measurements are shown in the main figures and the corresponding volume data is displayed as Supplementary Information.

Consistent with the effect of *raga-1* deletion mutants[12], ubiquitous AID of RAGA-1 reduced pharynx and body growth rates and extended larval stage durations (Fig. 3c, Supplementary Fig. 5a, b). Growth inhibition was quantitatively dependent on the IAA concentration applied. Tir1 has weak activity towards the AID tag even in the absence of IAA[46,47], such that growth was reduced by 10–21% (depending on the larval stage) compared to a strain not expressing Tir1 (Supplementary

Fig. 5b). Throughout this study, we therefore include a strain lacking Tir1 expression, in addition to omitting IAA as a negative control.

## Pharynx-specific inhibition of RagA/*raga-1* reduces growth, but retains pharynx-to-body size proportions

To ask how tissue-specific AID of RAGA-1 affects the size proportion of the pharynx relative to the rest of the body, we used two strains expressing Tir1 only in selected tissues[41,45] (Fig. 3b). First, we expressed Tir1 specifically in the pharyngeal muscle under the control of the *myo-2* promoter[41]. Second, we restricted Tir1 expression to hypodermal and seam cells using the *col-10* promoter[45] (collectively referred to as epidermis[48] from here onwards). In either of these strains, pharynx as

well as body growth were quantitatively reduced upon addition of IAA (Fig. 3c, Supplementary Fig. 5a, b). Thus, depletion of RAGA-1 in a single tissue reduces growth non-autonomously also in other tissues. Epidermal RAGA-1 AID reduced growth less strongly than pharyngeal RAGA-1 AID, which may be due to biological differences between these two tissues, or due to technical differences in the knock-down efficiency.

To quantify the non-autonomous response between pharynx and body growth, we plotted log-transformed length of body and pharynx at all four moults (M1 to M4) against each other (Fig. 3d). An equivalent analysis of volumes is shown in Supplementary Figures (Supplementary Fig. 5c, d). After log transformation, body and pharynx length were near linearly related, as is expected for two tissues with near exponential growth. We call this relationship the P-line (P for pharynx), where the slope $m = \frac{d\log(length_{pharynx})}{d\log(length_{body})}$ of the P-line corresponds to the ratio of the exponential growth rate of pharynx and body ($m = \frac{\mu_{pharynx}}{\mu_{body}}$). Analysis of the P-line under perturbed conditions distinguishes three different scenarios: (1) In the absence of tissue growth coordination, tissue-specific inhibition of growth is expected to change the ratio between pharynx and body growth rates and thus alter the slope $m$ of the P-line. (2) If unperturbed tissues respond with a proportional change in their growth rate the slope of the P-line remains constant. (3) A temporary deviation from the unperturbed organ growth proportions, followed by an appropriate adjustment of systemic growth rates would produce a P-line with unchanged slope, but a change in y-intercept.

In some experiments, we observed a very rapid length extension immediately after hatching (Fig. 3c). This rapid growth could be due to rapid biosynthesis immediately after hatching, due to an expansion unrelated to biosynthesis, or due to technical effects and measurement noise. To avoid confounding effects of this rapid size increase, we start our analysis at 30% passage of the L1 stage rather than immediately after hatching. Pharynx-to-body proportions upon epidermal and pharyngeal RAGA-1 AID were closest to scenario #3 described above. Without RAGA-1 AID, the slope $m$ of the P-line was 0.44. Under pharyngeal RAGA-1 AID, the P-line was shifted down by 5–8% but did not systematically change in slope (Fig. 3d), except for the highest IAA concentration at the last molt ($m = 0.43, 0.41, 0.43, 0.43, 0.37$ for IAA from 0 to 1 mM). Similarly, epidermal RAGA-1 AID caused a near parallel upshift of the P-line by less than 5% without systematically changing its slope ($m = 0.43, 0.45, 0.45, 0.45, 0.43$). Thus, in both tissues, RAGA-1 AID resulted in only a small difference in length proportions that plateaued at 5–8% and that did not scale with the degree of RAGA-1 inhibition (Fig. 3e, Supplementary Fig. 5d). This saturation of pharynx length deviation suggests that tissues in which RAGA-1 was not depleted by AID reduced their growth rate near proportionally to the tissue targeted by AID. Pharynx-to-body length proportions therefore remained nearly constant even under strong tissue-specific impairment of RAGA-1.

Analysis of volume instead of length led to equivalent conclusions (Supplementary Fig. 5c, d), with only minor quantitative differences. For pharyngeal depletion of RAGA-1, the maximal deviation of the pharynx volume was larger than for length (15% vs. 5%). For epidermal RAGA-1 AID, volume deviations were smaller than length deviations (−3% vs. +5%, Supplementary Fig. 5d). This difference between the effects on length and volume is explained by a slight change in the body length-to-width ratio under epidermal RAGA-1 AID (Supplementary Fig. 5e). We note that while pharynx-to-body proportions were retained (Fig. 3d, e), pharyngeal depletion of RAGA-1 reduced the overall length (Fig. 3c), indicating that experimental interference can change the body length without changing pharynx-to-body length scaling.

Repeating the same experiments using an AID-tagged allele of the insulin signalling receptor DAF-2 instead of RAGA-1 produced near identical results. Like AID of RAGA-1, pharyngeal length deviation plateaued at less than 5% deviation despite significant growth inhibition of the pharynx (Supplementary Fig. 6). The observed robustness in pharynx length proportions is thus not a specific response to RAGA-1 inhibition, but likely a more general response to an imbalance in tissue growth or size.

We conclude that pharynx-to-body length and volume proportions are robust to even strong tissue-specific inhibition of growth.

## Body growth has ultra-sensitive dependence on pharynx size

To quantify the relation between size and growth of pharynx and body, and to estimate how deviations from this relation would affect pharynx-to-body proportions, we compared the experimental data to the following mathematical model (Fig. 4a).

$$\frac{db}{dt} = b^* \mu_b^* \sigma_b^* f_b(d_p) \tag{1}$$

$$\frac{dp}{dt} = p^* \mu_p^* \sigma_p^* f_p(d_p) \tag{2}$$

$\mu_p$ and $\mu_b$ are the growth rates of pharynx size $p$ and body size $b$ in the absence of experimental perturbation. $\sigma_p$ and $\sigma_b$ represent the degree of experimental growth inhibition ($0 < s = \frac{\mu_{perturbed}}{\mu_{unperturbed}} < 1$, i.e. $s = 1$ corresponds to unperturbed growth). $d_p = \frac{p_{perturbed}}{p_{unperturbed}}$ corresponds to the relative deviation from the unperturbed pharynx size (P-line). $f_b$ and $f_p$ describe how body and pharynx growth relate to the pharyngeal size deviation $d_p$. Experimentally, $f_b(d_p)$ and $f_p(d_p)$ can be determined by plotting the measured deviation from the unperturbed pharynx length or volume $d_p$ against the observed reduction in body or pharynx growth rates (Fig. 4b, Supplementary Fig. 7b).

Length measurements from all IAA concentrations and larval stages were well approximated by a Hill function $f_b(d_p) = \frac{1}{1+\left(\frac{d_p}{k}\right)^n}$ with half-inhibition constant $k = 4.4\%$ and a Hill coefficient $n = 6.8$. The response of pharyngeal growth to epidermal growth inhibition $f_p(d_p) = \frac{1}{1+\left(\frac{d_p}{k}\right)^n}$ had a similar half-way point of about 5%, although the available experimental data did not allow a quantitative fit to the model (Supplementary Fig. 7a). Simulations of coupled body and pharynx growth using the fitted Hill function $f(d_p) = f_p(d_p) = f_b(d_p)$ produced length trajectories close to experimental observations (Fig. 4c, d). Thus, stage- and condition-specific deviations from the idealized Hill function (Fig. 4a, Supplementary Fig. 7a) do not substantially contribute to the observed invariance of pharynx-to-body length proportions.

Like for lengths, the relation between pharynx volume and the body growth rate was also well approximated by a Hill function (Supplementary Fig. 7b), albeit with slightly different parameters. In conclusion, under tissue-specific depletion of RAGA-1, the body growth rate has ultra-sensitive dependence on the deviations from the appropriate pharynx size and vice versa.

## Models that lack of ultra-sensitive coupling produce large deviations in pharynx length

Figure 4a shows that the relation between body growth and pharynx size follows a steep Hill function. We next asked how important this ultra-sensitive coupling was for retaining correct pharynx-to-body length proportions by investigating models with alternative expressions for $f_b(d_p)$ and $f_p(d_p)$. Specifically, we simulated independent growth of pharynx and body ($f(d_p) = 1$), which would be expected in

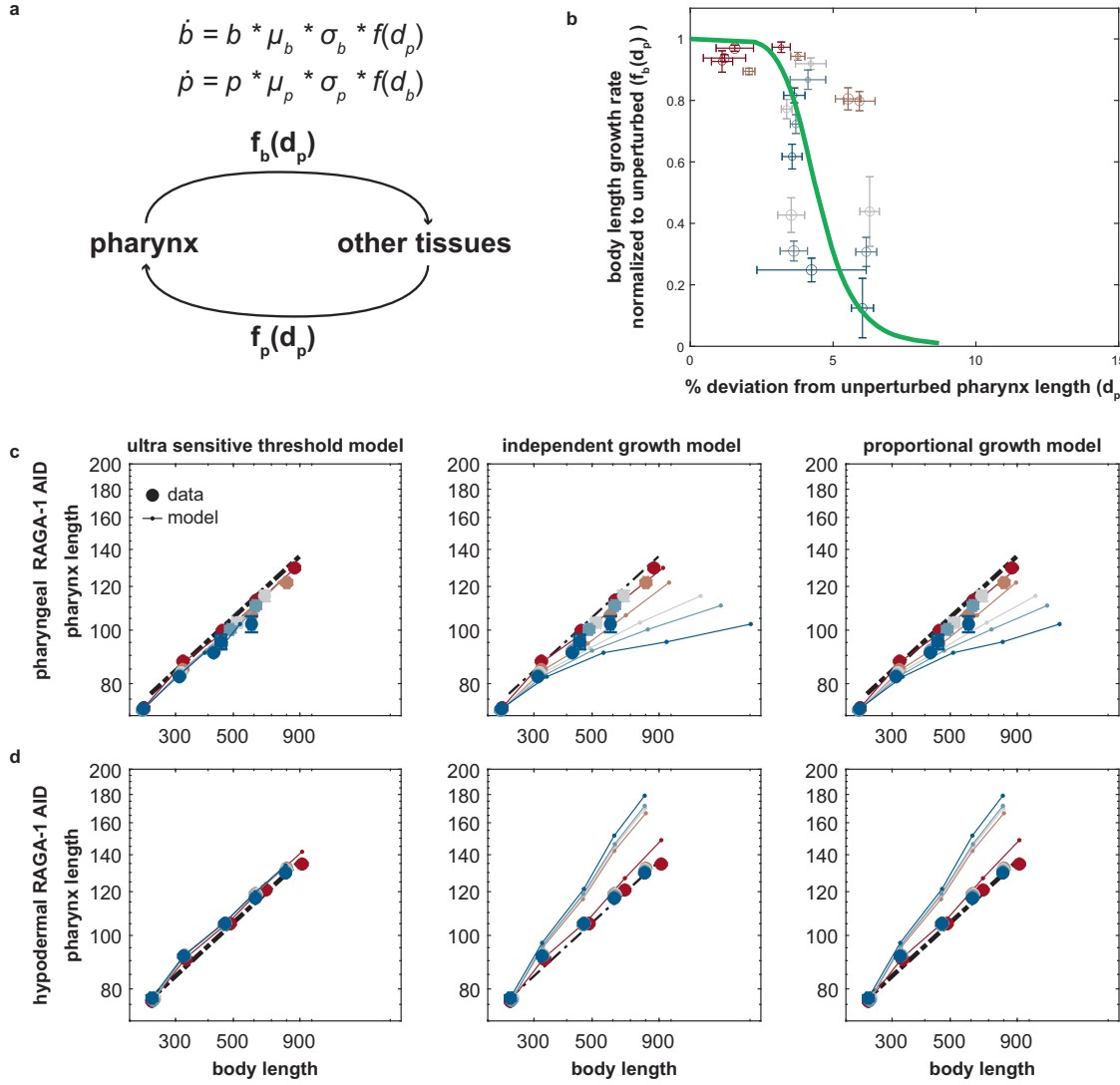

**Fig. 4 | An ultra-sensitive relation between pharynx size and body growth is required for robustness of pharynx-to-body length proportions to tissue-specific RAGA-1 inhibition. a** Schematic illustration of mathematical model as described in the text. **b** Body length growth rate (Δlog(length)/Δt) as a function of deviation of pharynx length $d_p$ from the P-line under pharyngeal growth inhibition. Growth is normalized to unperturbed growth. Color indicates IAA concentration increasing from red (0 mM) to blue (1 mM). Circle size indicates the larval stage (L1 being the smallest). Green line is a fitted Hill function as described in the main text. **c** Comparison of experimental data (solid circles) from pharyngeal RAGA-1 AID to three different models (thin lines) as described

in the main text with $f_p(d_p) = \frac{1}{1+(\frac{d_p}{P})^n}$ (left), $f(d_p) = 1$ (middle), $f(d_p) = 1 - d_p$ (right).

Colours as in (**a**). The ultra-sensitive threshold model is close to experimental data. The proportional and the independent models are inconsistent with the experimental data. **d** As (**c**), but for epidermal RAGA-1 AID. **b**–**d** error bars in vertical and horizontal direction indicate standard error of the mean between day-to-day repeats. Absence of error bars means that they were smaller than the marker size. Number individuals $n$, $26 < n < 259$ from number of day-to-day repeats $m$ between 2 and 10. See Supplementary Tables 1 and 2 for n and m of each condition.

the absence of any coupling. Second, we modelled proportional scaling of body growth to pharynx length ($f(d_p) = 1 - d_p$), which could, e.g., occur due to a proportional limitation of food uptake by a smaller pharynx. Both alternative models strongly deviated from the experimental observations (Fig. 4c, d) and were insufficient to explain the experimental observations. Together, these analyses suggest that proportional coupling of pharynx and body growth through a reduced food uptake by a smaller pharynx would be insufficient to ensure correct pharynx-to-body length proportions.

**YAP-1 is required for the robustness of pharynx size proportions to epidermal growth inhibition**

To identify molecules involved in the coupling of pharynx and body growth, we used RNAi and genetic mutations to systematically impair canonical growth regulatory pathways. Specifically, we targeted genes

associated with (i) transforming growth factor β (TGFβ)[49], (ii) insulin and insulin-like growth factor (IIS)[50], (iii) mTOR[51], and (iv) Yes-associated protein/YAP signalling[52]. For each gene, we exposed animals with a *raga-1-aid* tag expressing Tir1 in the epidermis or in the pharynx to RNAi and imaged growth of individual animals in micro chambers. In parallel, we used a strain lacking Tir1 expression as a control for each RNAi to distinguish genes involved in a response to imbalanced tissue growth from those that are more generally involved in controlling pharynx length. This systematic analysis revealed that the gene *yap-1* is required for retaining pharynx-to-body length proportions when RAGA-1 is depleted selectively in the epidermis (Fig. 5a, c). YAP-1 is the single *C. elegans* ortholog of the human transcriptional co-activators *Yes associated protein* (YAP) and the *Transcriptional Co-Activator with PDZ-Binding Motif* (TAZ)[52], which function as mechanotransducers downstream of the Hippo signalling pathway[27,28].

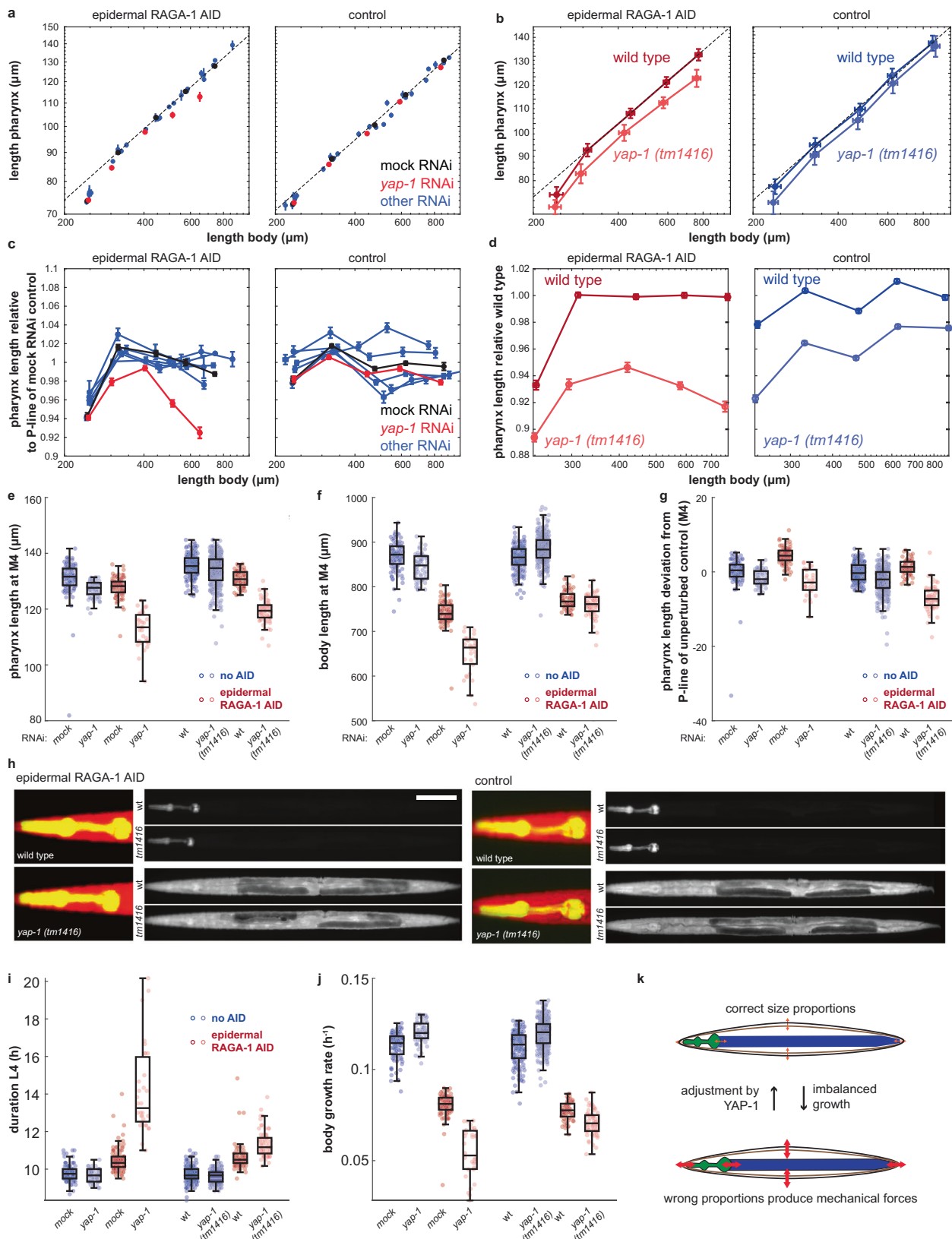

RNAi against other genes than *yap-1* had no, or a much weaker, impact on pharynx-to-body length proportions, although we cannot exclude insufficiency of knock-down for these genes. For example, we did not detect synergistic effects between RAGA-1 AID and RNAi of the TGFβ ligand *dbl*-1 or its downstream effector gene *lon-1* (Fig. 5a, c and Supplementary Fig. 8b). Similarly, a null mutation of the IIS effector

gene *daf-16* did not perturb pharynx-to-body growth coordination. Although *daf-16* mutants had a slightly reduced pharynx length under otherwise unperturbed conditions, this difference was not enhanced by tissue-specific RAGA-1 AID (Supplementary Fig. 8a). Notably, unlike for epidermal RAGA-1 AID, *yap-1* was not required to retain pharynx-to-body length proportions under pharyngeal AID of RAGA-1

**Fig. 5 | *yap-1* is required for robustness of pharynx-to-body length proportions towards epidermal RAGA-1 inhibition. a** Pharynx vs. body length at 30% of L1 stage and at moults M1 to M4 under epidermal RAGA-1 AID (left) and without Tir1 expression (right) and with indicated RNAi. Black: mock control, red: *yap-1*, blue: *dbl-1, lon-1, wts-1, ftt-2*. Black line: linear regression to M1 to M4 of mock RNAi. Error bars in x and y: SEM among individuals. If no error bars are visible, they are smaller than the marker. **b** As (**a**), but for wild type and *yap-1(tm1416)* on OP50-1. Error bars are SD among individuals. **c** As (**a**), but for deviation of pharynx length from unperturbed conditions (deviation from P-line of the respective mock RNAi). $p < 10^{-10}$ for comparison *yap-1(RNAi)* vs. mock RNAi for *col-10p:tir1* strain at M1 to M4 (ranksum test, two-sided). **d** As (**c**), but for wild type and *yap-1(tm1416)* on OP50-1. Error bars: SEM among individuals. **e** Box plot of pharynx length at M4. First 4 boxes from left: *yap-1(RNAi)* with controls; right: *yap-1(tm1416)* with controls. central line: median, box: interquartile ranges (IQR), whisker: ranges except extreme

outliers (>1.5*IQR), circles: individual data points. $p < 10^{-8}$ for interaction between RNAi or mutant and RAGA-1 AID (2-way ANOVA). **f** As (**e**), but for body length at M4. **g** As (**e**), but for deviation of pharynx length from P-line of corresponding negative control. **h** Representative images (within 1% of the median pharynx length) of indicated genotypes and treatments. Scale bar = 0.1 mm. **i** As (**e**), but for duration of L4 stage. **j** As (**e**), but for mean body growth rate during L4. **k** Speculative model for the role of *yap-1* in organ growth coordination: Attachment of the epidermis (brown) and the gastrointestinal tract (blue for intestine, green for pharynx) to the cuticle (black) causes forces (red) acting on cells of wrongly proportioned organs. As a mechanotransducer, YAP-1 may respond to these forces and induce an appropriate counteracting cellular growth response. **a–g, i, j** number of individuals are in Supplementary Table 2. Precise *p*-values and *p*-values for pairwise comparisons are in Supplementary Table 3.

(Supplementary Fig. 8b), suggesting tissue-specific differences in the mechanisms of organ growth coordination.

We confirmed the phenotypes of *yap-1(RNAi)* using the genetic allele *yap-1(tm1416)*[52], which deletes the region encoding the conserved WW domain (Supplementary Fig. 9a). Like *yap-1(RNAi)*, *yap-1(tm1416)* animals deviated from the wild type in their pharynx length (Fig. 5b, e, h) and in their pharynx-to-body scaling (Fig. 5d, g, h) specifically when exposed to epidermal RAGA-1 AID. The length deviations caused by *yap-1(tm1416)* were slightly weaker than those caused by *yap-1(RNAi)*, especially for total body length (Fig. 5f). This weaker phenotype of the mutant compared to the RNAi suggests that *yap-1(tm1416)* is not a complete null allele and retains partial activity. Alternatively, *yap-1(RNAi)* may have unspecific effects in addition to the role of *yap-1* in pharynx growth coordination revealed by the *tm1416* allele.

### Depletion of *yap-1* impairs larval development upon growth inhibition of the epidermis

To test the physiological importance of *yap-1* under tissue-specific depletion of RAGA-1, we quantified the rates of growth and development of *yap-1(RNAi) and yap-1(tm1416)* animals in micro chambers. Consistent with previous reports of only mild phenotypes of *yap-1* mutants[52], RNAi and mutation of *yap-1* alone did not slow down growth or delay larval stage progression (Fig. 5i, j). However, growth rate and developmental speed were reduced when *yap-1(RNAi)* or *yap-1(tm1416)* were combined with epidermal AID of RAGA-1 (Fig. 5i, j). Thus, *yap-1* is important to support normal growth rates specifically when combined with epidermal depletion of RAGA-1.

In addition to reduced growth, *yap-1(RNAi)*, but not *yap-1(tm1416)*, caused a partially penetrant larval arrest when combined with epidermal RAGA-1 AID (Supplementary Fig. 9b, c, e). Similar to the effect on body length (Fig. 5f, see above), this difference between *yap-1(RNAi)* and the *tm1416* allele may be due to remaining activity of the *tm1416* allele or due to unspecific RNAi. Supporting a specific effect of *yap-1(tm1416)* on larval stage progression, the *tm1416* allele enhanced the penetrance of larval arrest when combined with *yap-1(RNAi)* (Supplementary Fig. 9d).

In conclusion, mutation and knock-down of *yap-1* does not reduce the rate of organismal growth and the speed of larval development per se. However, *yap-1* is crucial for rapid growth and development, and for normal pharynx-to-body length proportions, when RAGA-1 is reduced in the epidermis.

## Discussion

Using time-lapse microscopy, we have observed a near perfect uniformity of the *C. elegans* pharynx volume and length with a coefficient of variation among individuals of less than 5%. The volume heterogeneity of the *C. elegans* pharynx is thus substantially smaller than the heterogeneity in total body volume[12] (Fig. 1). Indeed, unlike deviations in total body volume, pharyngeal volume deviations are corrected by

sub-exponential growth, such that large pharynxes undergo a smaller volume fold change than small pharynxes. Reminiscent of adders observed in bacteria, yeasts, and mammalian cells[7,9,53], the added volume per larval stage was near independent of the volume at the beginning of the larval stage (Fig. 2). However, despite these phenomenological parallels, uni-cellular and multi-cellular adders are likely mechanistically distinct. Unlike cellular adders, which function cell autonomously, pharyngeal growth is tightly coordinated with the growth of other tissues (Fig. 3).

The pharynx plays a crucially role in food uptake. It is thus likely that deviations in its size impact total body growth by changing the rate of nutrient uptake. However, our data suggests that there are additional mechanisms at play in maintaining robust pharynx-to-body proportions. First, endogenous fluctuations in the pharynx volume were not correlated to overall body growth, indicating that the pharynx volume does not restrict body growth within the endogenously observed range (Supplementary Fig. 2e). Second, body growth has an ultra-sensitive dependence on pharynx size with a 50% reduction of body growth occurring at a 5% reduction of pharynx length. Such a steep decline in growth is difficult to reconcile by food limitation due to a reduced pharynx size alone. Third, simulations showed that the ultra-sensitive coupling of pharynx and body growth is required for robust pharynx-to-body length proportions (Fig. 4). Fourth, growth coordination is bi-directional, occurring also when growth of the epidermis is inhibited by RAGA-1 AID, which does not reduce pharynx length (Fig. 4). Fifth, depletion of the gene *yap-1* impairs the robustness of pharynx-to-body proportions to tissue-specific growth inhibition by RAGA-1 AID (Fig. 5). This genetic dependence shows that molecular regulation is involved in maintaining correct length proportions of pharynx and body, in addition to any role played by limitation of food uptake.

The YAP/Hippo signaling pathway was initially identified by genetic mutants that hyperactivate YAP, causing increased imaginal disc size in *Drosophila*[29–33] and an enlarged liver in mice[34,35]. However, it has been questioned if YAP is indeed an immediate driver of cellular growth during normal development since loss-of-function mutations of YAP do not impair the growth and size of most tissues[36]. Consistently, we find that mutation and knock-down of *yap-1* alone does not reduce the larval body growth rate of *C. elegans* (Fig. 5i, j), although it does play an important role in cell polarity[54,55] and aging[52,56] at later stages. Whereas we found that *yap-1* is not required for rapid larval growth, it is essential for development of correct pharynx-to-body length proportions in the face of spatial imbalance in mTORC1 signalling (Fig. 5a–h). Our results may thereby resolve the discrepancy between the loss- and gain-of-function phenotypes of YAP observed in other systems, revealing a role of *yap-1* in organ growth coordination, albeit not in the stimulation of organismal growth per se. In future research, it will be important to address if YAP-1 globally senses growth or size imbalances, or if its response is mediated by direct crosstalk between the mTOR and the Hippo pathway[57].

In our experiments, RNAi of the upstream regulators in the Hippo pathway did not impair pharynx-to-body length proportions (Fig. 5a, c). Since the Hippo pathway is an inhibitor of *yap-1* activity[52], knockdown of upstream regulators is indeed not expected to phenocopy loss of *yap-1*, but rather mimic *yap-1* hyperactivation. Alternatively, *yap-1* may play a role independent of the Hippo pathway in *C. elegans*.

How could *yap-1* ensure appropriate pharynx-to-body size proportions? Given the conserved role of YAP-1 as a mechanotransducer[58], an attractive hypothesis is that marginal disproportions in tissue size produce pulling forces among cells, or between cells and the cuticle, which trigger the activation of YAP-1 (Fig. 5k). YAP-1 may then trigger cell autonomous transcriptional responses that counteract this mechanical strain, or trigger the release of systemic or hormonal signals, as has been suggested for flies[59]. An important open question made accessible by this study is how cells quantitatively convert mechanical forces to an appropriate gene expression response that ensures the faithful organismal development to the correct size proportions.

## Methods

### Strains used in this study
All strains used in this study were created by genetic crosses from the following previously published and extensively outcrossed alleles: *bqSi577*[60], *wbmIs88*[61], *ieSi60*[41], *daf-2(bch40)*[62], *raga-1(wbm40)*[44], *reSi1*[45], *xeSi376*[41], *daf-16(mu86)*[63], *yap-1(tm1416)*[52]. See Supplementary Table 4 for a complete list of strains used in this study.

### Micro chamber preparation
Micro chambers were manufactured as described[12] using a PDMS-based master mould to produce wells in a 4.5% Agarose gel dissolved in S-basal. For all experiments, chamber dimensions were $600 \times 600 \times 20 \, \mu m$. As a food source, the bacterial strain OP50-1 was grown on NGM plates by standard methods, scraped off using a piece of 3% NGM agar without cholesterol and then filled into the wells of the agarose gel. Wells were filled with eggs at 2-fold stage and subsequently inverted onto a dish of 3.5 cm diameter with a high optical quality gas-permeable polymer bottom (ibidi). The remaining surface of the dish was covered with 3% low melting temperature agarose dissolved in S-basal (cooled down to below 42 °C prior to application). The agarose was overlayed with ~0.5 ml polydimethylsiloxane (PDMS) and the dish was sealed with parafilm to minimize water evaporation. PDMS was allowed to cure at room temperature on the microscope during the acquisition. Using a custom-made plate holder, six dishes could be imaged simultaneously on one microscope. Auxin (IAA, Sigma) solutions were freshly prepared on the day of the experiment as a 400x stock in EtOH and subsequently diluted to the indicated concentration in agarose to a final EtOH concentration of 0.25% immediately prior to use for micro chamber assembly.

### Imaging
All experiments were performed on a Nikon Ti2 epifluorescence microscope using a 10x objective with NA = 0.45 and a Hamamatsu ORCA Flash 4 sCMOS camera with a pixel size of 6.5 μm, leading to an effective pixel size of 650 nm in the sample plane. Temperature was maintained at 25 +/− 0.1 °C by an incubator enclosing the entire microscope and a feedback-controlled temperature regulator (ICE cube, life imaging services). Separately triggerable LEDs (SpectraX, lumencore) for 470 nm (GFP) and for 575 nm (wrmScarlet) were used as an excitation light source that was TTL-triggered to ensure rapid switching between wavelengths and imaging of red and green fluorescence within 10 ms. Acquisition times were kept below 10 ms, such that two color imaging was completed within 30 ms. Rapid image registration was crucial to minimize a shift between fluorescent channels in the absence of physical or chemical immobilization of the animals. Software-based autofocus of Nikon's NIS Elements software was used every 10 min using 575 nm excitation at low intensity. We confirmed that these acquisition settings did not impair growth, development, and fertility of the animals.

### Image analysis and calculation of volumes and growth rates
The outline of the worm body was determined by an edge detection-based algorithm, implemented in Matlab, as previously described[12]. The pharynx outline was determined by a pixel classifier trained using Ilastik[64]. Segmentations of body and pharynx were subsequently straightened using the body outline as a template as previously described[12]. Briefly, the midline of the worm was determined by iterative erosion, a spline was fitted to this midline, and a straightened image was created by assembling orthogonal cross-sections along the midline spaced at a distance of 1 pixel along the spline. Pharynx and body length were computed as the length of the midline. Volumes were inferred assuming rotational symmetry from straightened animals[38–40]. To this end, we summed up the volume of each midline-orthogonal cross-section, which was calculated by $\pi * (d/2)^2 * w$. $d$ is the width (diameter) of the straightened worm at the respective cross-section, $w$ is the spacing of cross-sections (1 pixel)[37]. A decision tree-based classifier[12] was used to determine the time point of hatching, and to identify time points where straightening failed (e.g. due to self-touching animals). Detection of moults and computation of growth rates was conducted as previously described from volume trajectories[12]. Volumes at hatch and larval molts were determined by fitting a linear model to 10 neighboring points of the log transformed volumes. Each volume trajectory was visually inspected and hatch/molt time and volume annotations were corrected manually if needed.

### Estimation of net instrument noise
Three independent approaches were taken to estimate the net instrument noise of pharynx and body volume and length. First, standard error of the regression to 10 neighboring point used to estimate volumes at hatch and molts revealed a precision of measurement for hatch to M4 for the pharynx of 4.8%, 1.9%, 1.6%, 1.6%, 1.3% and for the body of 1.2%, 0.96%, 0.82%, 0.70%, 0.87%. Second, volume trajectories were determined for 60 individuals at 5 min time resolution and split into two complementary sets, each of which had 10 min time resolution. The mean difference between the two measurement was close to the standard error of the linear regression. Third, an experiment was conducted, in which for 14 animals were imaged 20 times every 30 min. For each of the 20 images a separate autofocus was performed. The measurement error was determined as the median CV among repeated measurements at each time point.

### Comparison of observed growth to randomized simulations of adder and folder models
Simulations were performed as described[12] and started with the measured distribution of pharynx volume at M1 or after hatching. For simulations of an uncoupled folder, each individual was assigned a pharyngeal growth rate and a larval stage duration that were randomly and independently drawn from the experimentally measured distributions. For a coupled folder, each individual was assigned a volume fold change randomly drawn from the measured data. For simulations of an adder, each individual was assigned an added pharyngeal volume drawn from the measured data. From these randomly selected parameters the pharyngeal volume of the next larval stage was computed, and the process was iterated until the end of L4. Each randomization was repeated 1000 times for each day-to-day repeat and pooled for visualization. To exclude effects of large heterogeneity and technical noise in pharyngeal volume at hatch, two separate randomizations were conducted, one starting at hatch and one starting at M1. Both simulations yielded consistent results.

## Determination of trendlines in of ΔV vs. $V_1$ and comparison to simulations with technical noise

Trendlines in scatter plots were computed using robust linear regression using the robustfit() method of Matlab (v2021b) and default parameters.

To determine the impact of instrument noise, the CV of the biological heterogeneity was estimated by var(biological) = var(total) − var(technical). Starting volumes $V_{1, biol}$ were then drawn from a normal distribution with standard deviation corresponding to the biological heterogeneity and sample size corresponding to the number of individuals measured experimentally for the respective larval stage. A folder was simulated by multiplying with a fold-change $FC_{biol}$ drawn from a normal distribution with standard deviation corresponding to the observed heterogeneity to yield a volume distribution $V_{2, biol}$. Subsequently, technical error was added to $V_{1,biol}$ and $V_{2,biol}$, drawn from a normal distribution with standard deviation of the technical noise determined for the respective larval stage, yielding simulated $V_{1,biol + tech}$ and $V_{2,biol + tech}$, from which $\Delta V_{biol + tech} = V_{2, biol + tech} - V_{1, biol + tech}$ was computed. $V_{1,biol + tech}$ and $\Delta V_{biol + tech}$ were normalized to their respective means, equivalent to the treatment of experimental data in Fig. 3, and the slope of this relation was determined by linear regression. This simulation was conducted 10'000 times to compare simulations to the experimentally observed relation between with ΔV and $V_1$.

### Correction of regression slopes for attenuation bias

The slope of a linear regression between two observed variables $V_{1, observed}$ and $V_{2, observed}$ is biased towards zero due to measurement error in $V_1$. To correct for this bias (called least squares attenuation bias), we computed the reliability ratio $\lambda = \frac{\text{var}(V_{1,biol})}{\text{var}(V_{1,biol}) + \text{var}(V_{1,tech})}$, where var($V_{1,biol}$) is the variance of $V_1$, corrected for the contribution of technical error: $\text{var}(V_{1,biol}) = \text{var}(V_{1,observed}) - \text{var}(V_{1,tech})$. var($V_{1,tech}$) is the experimentally determined measurement error. The measured slopes between $V_1$ and $V_2$ were for then corrected for the attenuation bias by: $\text{slope}_{corrected} = \text{slope}_{measured} / \lambda$.

### RNAi

All RNAi experiments were conducted by feeding using HTT115 clones retrieved from available libraries[65,66] and validated by Sanger sequencing. (clone numbers: *dbl-1*: sjj_T25F10.2; *lon-1*: sjj_F48E8.1, *yap-1*: sjj_F13E6.4, *ftt-2*: sjj_F52D10.3, *wts-1*: sjj_T20F10.1, *rsks-1*: sjj2_Y47D3A.16). RNAi was initiated on plates one generation prior to loading eggs in micro chambers. For RNAi plate preparation, bacterial clones were grown to saturation over night in LB with Ampicillin (100 μg/ml) and dispensed on NGM plates containing 4 mM IPTG and 100 μg/ml Carbenicillin. L4 stage animals grown on OP50-1 were transferred to RNAi plates and their embryonic progeny was transferred into micro chambers containing bacteria expressing the same RNAi clone using an eyelash glued to a pipette tip. For RNAi inside micro chambers, bacterial over night cultures were induced in liquid for 4 h using 4 mM IPTG, concentrated by centrifugation, and dispensed onto NGM plates to absorb the remaining liquid. After drying for 1 h, bacteria were scraped off the plate and used as described above for OP50-1.

### Averaging of growth rates and volumes after re-scaling

To average trajectories of growth rates, volume, and length without perturbing alignment of moults individual trajectories were re-scaled to the larval stage duration and averaged after interpolation at 100 points per larval stage. Data was then averaged over all individuals by taking the median at each of the 100 points per larval stage, and the averaged data was rescaled to the average larval stage duration.

### Computation of width and growth rates

Unless indicated otherwise, average growth rates per larval stage were calculated as the difference between the natural logarithm of the volume at the beginning and the end of the larval stage divided by the larval stage duration ($\mu = \Delta \log(V)/\Delta t$). Average width ($w$) of body and pharynx was calculated from volume $v$ and length $l$ as follows: $w = 2\sqrt{v/(\pi l)}$.

### Reporting summary

Further information on research design is available in the Nature Portfolio Reporting Summary linked to this article.

## Data availability

The volume and growth rate data generated in this study are provided as source data to the figures. The raw images are too large to be shared via available repositories and available from the corresponding author upon request. Source data are provided with this paper.

## Code availability

Code for image analysis was previously published and is available here: https://github.com/btowbin/NatComm2022 (https://doi.org/10.5281/zenodo.10032867). Code for simulations is provided in source data of the corresponding figures. Matlab (v2021b) was used to execute the code.

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

## Acknowledgements

We are thankful to Cihan Elci for technical assistance and to Rutger Hermsen, Helge Grosshans, Stephan Peischl, and Rafael Sauter for useful discussions and helpful feedback on the manuscript. We acknowledge support by the Microscopy Imaging Center at the University of Bern. This work received funding from the Swiss National Science Foundation (SNSF) in the form of an Eccellenza Professorial Fellowship (PCEFP3_181204) to B.D.T., the Novartis Foundation for Medical-Biological Research (Grant #20A011), and the Berne University Research Foundation. P.L. received funding from an SNSF Swiss Postdoctoral Fellowships (TMPFP3_209681). This work was funded by R01AG059595 (W.B.M.) and R01AG044346 (W.B.M.). Some strains were provided by the CGC, which is funded by NIH Office of Research Infrastructure Programs (P40 OD010440).

## Author contributions

B.D.T. conceived the study and wrote the manuscript. K.S., B.D.T, I.G., P.L., and W.M.B. edited the manuscript. K.S., B.D.T., I.G., and P.L. conducted the experiments and performed computational analysis. A.L. and W.B.M. designed, created, and validated the *raga-1-aid* allele.

## Competing interests

The authors declare no competing interests.
