## [Peer Review File · Nature Communications]

Maintenance of appropriate size scaling of the *C. elegans* pharynx by YAP-1REVIEWER COMMENTS

Reviewer #1 (Remarks to the Author):

Ultra-sensitive coupling between organ growth and size by YAP-1 1 ensures uniform body plan proportions in *C. elegans*

Stojanovski et al.

This work investigates the coupling between pharynx and body growth in *C. elegans*. Using previously established high throughput fluorescence imaging, and image processing, the authors quantify pharynx and body length (and volume) and find a nice scaling across developmental stages. The authors use an auxin inducible system to try to autonomously perturb pharynx size and hypodermal size. They observe that a similar scaling holds under these perturbations. A minimal model is used to demonstrate that body volume is ultrasensitively coupled to the pharynx size. Finally, the authors show that inhibition of yap-1 (Yes-associated protein/YAP) via RNAi results in impaired larval development – but only when the growth of the epidermis is inhibited. I believe this perhaps one of the first papers to measure scaling in a multicellular organisms by making careful measurements of organ and body size.

Here are some comments that need to be addressed before publication:

I am uncomfortable with the generalisation that pharynx is the same as all organs. While the authors are careful in some places, I think it would be more prudent not to generalise the results to all organs (including in the title). There are many reasons to believe other organs may not be similarly linked to body size.

One important comment regarding the organisation and the argument of the paper regarding ultra sensitivity is the way the model is presented. By looking at the data (fig 4b), that is fit to the Hill function (with a Hill coefficient of 6!), isn't it obvious that $f(dp)$ cannot be 1, or $1 - dp$? The authors state in line 243 that dp is defined by a hill function. The model is laid out further down in the paper, and other functional forms of dp are tested. I find the argument a bit circular. Some reorganisation or further justification is required.

There is lot of literature that discusses size scaling, and I was surprised to see that the authors have not really touched on that topic, not even in the introduction. This might make a nice addition to motivate the paper.

Some minor comments:

Wherever applicable can authors use volume instead of size?

Line 91 - When defining μ_{abs} - is this $(dV/dt)/V$ – can it be defined in those terms.

Line 116 “We distinguished three different scenarios” – this referring to previous work – I have seen this work and they define it as adder, size, and folder – it is not labelled the same way in this work.

Line 122 – There’s a huge jump going from description of simulations which point to an adder mechanism for the pharynx, to talking about the coupling between the organ and the whole body. How does the simulation account for this coupling? Some explanation is lacking here.

Line 182 – lines are referred as epidermal and hypodermal interchangeably in the rest of the paper – please rectify for consistency.

Line 193 – Please mention the actual value of the slope (m) – this would give an idea of the actual scaling

Line 201 – I did not understand the basis for this statement – how do you know the length extension is independent of biomass?

Line 215 – “We note that while pharynx-to-body proportions were retained (Fig. 3d, e), pharyngeal depletion of RAGA-1 reduced the overall size (Fig. 3c), indicating that total body size and pharynx-to-body proportions are controlled by separable mechanisms.” Again, the basis of this statement is not clear from the previous section. Please clarify the writing.

Line 303 – “In conclusion, yap-1 is dispensable for growth and development of *C. elegans* per se, but its activity is crucial for the robustness of pharynx-to-body size proportions under growth imbalance between tissues.” Many assumptions are being made here. Does one know that yap-1 doesn’t have some sort of pleiotropic effect in later larval stages?

Comments on figures:

Fig 1 a

Are you really measuring egg pharynx size?

Can authors speak to the resolution they can obtain on their platform with really small pharynx. Mentioned briefly in line 108 – but not enough is said.

Fig 3 d.

Can the legend be improved – what are circles, squared etc? colour code is fine as it is consistent across – representing IAA concentration.

Reviewer #2 (Remarks to the Author):

This manuscript combines highly quantitative experiments with modelling to reveal a ultra-sensitive coupling between organ size and growth rate of the body to maintain body proportions. The work is thoughtful, detailed, and provides an important advance.

Major issues:

1. Noise Estimates and Instrument Noise

There are several aspects in the quantification that require clarification in the methods. To show that biological noise estimates are accurate, it is important to show that the noise from non-biological sources are negligible or do not impact the conclusions. Since these issues are technical, the authors should address the questions regarding the set of related points below in the methods.

a. What is the pixel size in microns? This parameter limits the accuracy of the size measurements. Does it contribute disproportionately to the CV of length and volume measurements greater in smaller animals?

b. What is the net instrument noise due to pixel size, autofocusing, segmentation, and computationally straightening worms? What is their net contribution to the CV?

c. How are the errors propagated in the estimation of volume and does that account for the approximately 10-fold difference in the observed CV of pharynx length vs volume in Fig 1d vs Supplemental Fig 1c? Is there validation by comparing the estimates from taking multiple optical sections? While volume estimates are useful, the same points can be made with length data in Fig 1 with fewer caveats. Standardising on length would improve consistency

across the entire manuscript.

2. Role of yap-1

There are some caveats to the analysis of yap-1 and potentially other models for yap-1's role.

a. The role of yap-1 is depicted as a major point in the manuscript based on the prominence in the title and abstract. However, the evidence for the role of yap-1 is solely based on RNAi, which may not be specific. It is important to validate this result with an outcrossed yap-1 mutant. Alternatively, the authors can be explicit about the caveats and reduce the prominence of yap-1 in the manuscript.

b. To justify the title that yap-1 mediates ultra-sensitive coupling would require additional experiments to show that the Hill co-efficient is reduced when yap-1 activity is disrupted. Alternatively, this point can be addressed by revising the title to downplay this specific claim, which would in no way reduce the importance and interest of this work.

c. yap-1 signalling is presented as a general mechanism for robustness of proportionate growth. Is it not just as likely that yap-1 is specific to the raga-1 pathway as there is literature relating mTOR signalling to Hippo/YAP (e.g., Csibi & Blenis, Nature Cell Biology 2012)? These alternative models should be mentioned in the discussion.

Minor issues:

Does raga-1 affect RNAi efficacy? The authors suggest that yap-1 only comes into play when there is raga-1 perturbation. But there is no evidence provided to rule out the possibility that the RNAi is more severe under raga-1 loss. The point that other RNAi treatments do not synergize with raga-1 does not address this issue since those RNAi treatments may be true negatives. This caveat should be stated in the results section.

Introducing sizing mechanisms (e.g., adder, folder) in greater detail would provide context to make it easier for readers to understand and appreciate many aspects of this work.

Fig 2 could benefit from a color scale bar for a-b and c.

Fig 3e – the sizes of the circles are obscured by the error bars. A larger panel could address this visualization issue.

Typo in Fig 2 legend: "...produces and adder-like..."

Typo in Supplemental Fig 2c-d y-axis label: "voulme"

REVIEWER COMMENTS

Reviewer #1 (Remarks to the Author):

Ultra-sensitive coupling between organ growth and size by YAP-1 1 ensures uniform body plan proportions in *C. elegans*, Stojanovski et al.

This work investigates the coupling between pharynx and body growth in *C. elegans*. Using previously established high throughput fluorescence imaging, and image processing, the authors quantify pharynx and body length (and volume) and find a nice scaling across developmental stages. The authors use an auxin inducible system to try to autonomously perturb pharynx size and hypodermal size. They observe that a similar scaling holds under these perturbations. A minimal model is used to demonstrate that body volume is ultrasensitively coupled to the pharynx size. Finally, the authors show that inhibition of yap-1 (Yes-associated protein/YAP) via RNAi results in impaired larval development – but only when the growth of the epidermis is inhibited. I believe this perhaps one of the first papers to measure scaling in a multicellular organisms by making careful measurements of organ and body size.

We thank the reviewer for endorsing the novelty of our work.

Here are some comments that need to be addressed before publication:

I am uncomfortable with the generalisation that pharynx is the same as all organs. While the authors are careful in some places, I think it would be more prudent not to generalise the results to all organs (including in the title). There are many reasons to believe other organs may not be similarly linked to body size.

We agree with the reviewer and have revised the text accordingly, including in the title, which now reads:

Maintenance of appropriate size scaling of the *C. elegans* pharynx by YAP-1

One important comment regarding the organisation and the argument of the paper regarding ultra sensitivity is the way the model is presented. By looking at the data (fig 4b), that is fit to the Hill function (with a Hill coefficient of 6!), isn't it obvious that $f(d_p)$ cannot be 1, or $1 - d_p$? The authors state in line 243 that d_p is defined by a hill function. The model is laid out further down in the paper, and other functional forms of d_p are tested. I find the argument a bit circular. Some reorganisation or further justification is required.

We are thankful for this comment, which allowed to better explain the purpose of comparing different forms of $f(d_p)$. The purpose of simulating alternative models is indeed not to validate that $f(d_p)$ is a Hill function. Instead, we do this to evaluate the consequence of a deviation from said Hill function on pharynx-to-body length proportions. We believe this comparison is of interest to the reader, as it provides support that maintenance of pharynx-to-body proportions is not a "passive" consequence of food uptake restriction due to a smaller pharynx. We explain this reasoning in the revised text (In 282ff.), which now reads:

Models that lack of ultra-sensitive coupling produce large deviations in pharynx length

Fig. 4a shows that the relation between body growth and pharynx size follows a steep Hill function. We next asked how important is this ultra-sensitive coupling was for retaining correct pharynx-to-body length proportions by investigating models with alternative expressions for $f_b(d_p)$ and $f_p(d_p)$. Specifically, we simulated independent growth of pharynx and body ($f(d_p) = 1$), which would be expected in the absence of any coupling. Second, we modelled proportional scaling of body growth to pharynx length ($f(d_p) = 1 - d_p$), which could, e.g., occur due to a proportional limitation of food uptake by a smaller pharynx. Both alternative models strongly deviated from the experimental

observations (Fig. 4c, d) and were insufficient to explain the experimental observations. Together, these analyses suggest that proportional coupling of pharynx and body growth through a reduced food uptake by a smaller pharynx would be insufficient to ensure correct pharynx-to-body length proportions.

There is lot of literature that discusses size scaling, and I was surprised to see that the authors have not really touched on that topic, not even in the introduction. This might make a nice addition to motivate the paper.

We agree and now motivate our study by the existing literature on size scaling in the introduction, which now reads (ln. 48ff):

At the scale of organs, size proportions follow robust allometric relations^{1,2}, and proportions of body parts usually scale appropriately over wide range of body sizes³. Combined experimental and theoretical work provides elegant explanations for how tissue patterning appropriately scales with tissue size during development^{4,5}. For example, a negative feedback between two diffusible components can ensure scale invariance of morphogen gradients³⁻⁶. Genetic experiments revealed tissue autonomous and systemic mechanisms that control organ size⁷. For instance, morphogen gradients are thought to limit the lateral expansion of imaginal discs in *Drosophila melanogaster*^{8,9} and damaged imaginal discs trigger systemic responses via secretion of the relaxin-like signalling peptide *Dilp8*^{10,11}. Similarly, unilateral inhibition of limb growth in mice triggers a growth response that retains proper limb symmetry^{12,13}. However, individual organ growth trajectories have rarely been measured *in vivo* over time, and how deviations in organ size are dynamically corrected during development remains poorly understood.

Some minor comments:

Wherever applicable can authors use volume instead of size?

We now specify throughout the manuscript whether we talk about volume or length. In some instances, we refer to a both, length and volume. In these cases, we use the term size to avoid repeated use of lengthy wording of "volume and length".

Line 91 - When defining μ_{abs} - is this $(dV/dt)/V$ - can it be defined in those terms.

μ_{abs} stands for (dV/dt) , μ stands for $(dV/dt)/V = d \log(V)/dt$ This is now specified more clearly in the revised text in ln. 98ff.

Throughout this article, growth rate refers to the change in log transformed volume per time ($\mu = d \log(vol) / dt = (d vol/dt)/vol$), i.e., the growth rate normalized to the current size, unless specified as the absolute growth rate ($\mu_{abs} = d vol / dt$), which indicates the absolute change in volume per time.

Line 116 "We distinguished three different scenarios" - this referring to previous work - I have seen this work and they define it as adder, size, and folder - it is not labelled the same way in this work.

This appears to be a misunderstanding. In our previous work (Stojanovski et al. (2022)¹⁴), we explored the same models as in this current study: adder, coupled folder, and uncoupled folder. In Stojanovski et al (2022)¹⁴, we additionally analyzed a combined model for body volume, where L1

was an adder, and L2-L4 was a folder. This analysis, however, is not relevant in the case of the pharynx. Instead, we simulate the pharynx as the three types of models either starting from M1 (shown in main figure) or starting from hatch (in supplemental figures). The reason for starting simulations for the pharynx at M1 in the main text is to exclude confounding impact of higher measurement noise, or other L1-specific effects, for pharyngeal volumes at hatch.

Line 122 – There's a huge jump going from description of simulations which point to an adder mechanism for the pharynx, to talking about the coupling between the organ and the whole body. How does the simulation account for this coupling? Some explanation is lacking here.

We now clarify that the analysis of folder vs. adder does not consider specific mechanisms, such as coupling, and is purely phenomenological (as is also the case for other experimental systems where adders have been observed). The simulations shown in Figure 1 do therefore indeed not account for the coupling. Importantly the work shown in Figures 3,4 and 5 goes beyond the phenomenological description of an adder and presents mechanistic insight in how pharyngeal size scaling is achieved. The corresponding section now reads (175ff):

The adder-like behavior of the pharynx suggests that growth is controlled in a size-dependent manner to ensure pharyngeal size uniformity among individuals but does not inform on the mechanism underlying this control. At least two distinct, not mutually exclusive, classes of mechanisms are conceivable. First, the narrow pharyngeal volume distribution could be due to precise, tissue-autonomous control. Second, pharyngeal volume scaling could involve crosstalk with other tissues. To distinguish between these two scenarios, we developed an experimental approach perturb pharynx growth and other tissues by tissue-specific auxin-induced degradation (AID)¹⁵ of the mTORC1 activator RagA/RAGA-1¹⁶⁻¹⁸ (Fig. 3a).

Line 182 – lines are referred as epidermal and hypodermal interchangeably in the rest of the paper – please rectify for consistency.

Indeed, we mistakenly used "hypodermal" in the figures and "epidermal" in the text, which is now corrected. The revised text introduces the term epidermal = hypodermal + seam cells once in line 204, and subsequently uses the term epidermal to describe the cells where the *col-10* promoter is expressed.

Line 193 – Please mention the actual value of the slope (m) – this would give an idea of the actual scaling.

The values of the slopes are now added to the text (ln. 229ff):

Without RAGA-1 AID, the slope m of the P-line was 0.44. Under pharyngeal RAGA-1 AID, the P-line was shifted down by 5-8% but did not systematically change in slope (Fig. 3d), except for the highest IAA concentration at the last molt ($m = 0.43, 0.41, 0.43, 0.43, 0.37$ for IAA from 0 to 1000 μM). Similarly, epidermal RAGA-1 AID caused a near parallel upshift of the P-line by less than 5% without systematically changing its slope systematically ($m = 0.43, 0.45, 0.45, 0.45, 0.43$).

Line 201 – I did not understand the basis for this statement – how do you know the length extension is independent of biomass?

We agree and revised the text accordingly (ln. 225):

This rapid growth could be due to rapid biosynthesis immediately after hatching, due to an expansion unrelated to biosynthesis, or due to technical effects and measurement noise.

Line 215 – “We note that while pharynx-to-body proportions were retained (Fig. 3d, e), pharyngeal depletion of RAGA-1 reduced the overall size (Fig. 3c), indicating that total body size and pharynx-to-body proportions are controlled by separable mechanisms.” Again, the basis of this statement is not clear from the previous section. Please clarify the writing.

Agreed and changed to (In. 245):

We note that while pharynx-to-body proportions were retained (Fig. 3d, e), pharyngeal depletion of RAGA-1 reduced the overall length (Fig. 3c), indicating that experimental interference can change body length without changing pharynx-to-body length scaling.

Line 303 –“In conclusion, *yap-1* is dispensable for growth and development of *C. elegans* per se, but its activity is crucial for the robustness of pharynx-to-body size proportions under growth imbalance between tissues.” Many assumptions are being made here. Does one know that *yap-1* doesn't have some sort of pleiotropic effect in later larval stages?

Agreed and changed to (In. 339):

In conclusion, mutation and knock-down of *yap-1* does not reduce the rate of organismal growth and the speed of larval development *per se*. However, *yap-1* is crucial for rapid growth and development, and for normal pharynx-to-body length proportions, when RAGA-1 is reduced in the epidermis.

Pleiotropic effects in later larval stages are now mentioned in the discussion (In. 374):

Consistently, we find that mutation and knock-down of *yap-1* alone does not reduce the larval body growth rate of *C. elegans* (Fig. 5i, j), although it does play an important role in cell polarity^{19,20} and aging^{21,22} at later stages.

Comments on figures:

Fig 1 a: Are you really measuring egg pharynx size? Can authors speak to the resolution they can obtain on their platform with really small pharynx. Mentioned briefly in line 108 – but not enough is said.

We are measuring pharynx size immediately after hatching, not in the egg (the first time point of larval development). As shown in the new Supplemental Figure S2, our measurements of pharynx volume and length are precise even in these small animals (~5% error) and precision further improves at later stages (~2%).

We now extensively discuss the implications of measurement noise on our conclusions. Overall, the impact of the measurement noise is very small. See also our answer to reviewer #2.

Fig 3 d. Can the legend be improved – what are circles, squared etc? colour code is fine as it is consistent across – representing IAA concentration.

The visualization is now improved. The apparent squares in the previous figure were due to the cap of the error bars. We now show error bars in black and without cap, such that circles are visible more clearly. Where invisible, the error bar is smaller than the marker.

Reviewer #2 (Remarks to the Author):

This manuscript combines highly quantitative experiments with modelling to reveal a ultra-sensitive coupling between organ size and growth rate of the body to maintain body proportions. The work is thoughtful, detailed, and provides an important advance.

We thank the reviewer for the endorsement of the quality and impact of our work.

Major issues:

1. Noise Estimates and Instrument Noise

There are several aspects in the quantification that require clarification in the methods. To show that biological noise estimates are accurate, it is important to show that the noise from non-biological sources are negligible or do not impact the conclusions. Since these issues are technical, the authors should address the questions regarding the set of related points below in the methods.

We now provide quantifications of the noise from non-biological sources and show that this noise does not impact our conclusions (see detailed answers below).

a. What is the pixel size in microns? This parameter limits the accuracy of the size measurements. Does it contribute disproportionately to the CV of length and volume measurements greater in smaller animals?

We now state the pixel size in the methods (ln. 418):

All experiments were performed on a Nikon Ti2 epifluorescence microscope using a 10x objective with NA=0.45 and a Hamamatsu ORCA Flash 4 sCMOS camera with a pixel size of 6.5 μm, leading to an effective pixel size of 650 nm in the sample plane.

Note that the pixel size is close to the diffraction limit according to Abbe's definition: $r = 2 \cdot \lambda / (2 \text{ NA}) = 571 \text{ nm}$.

Below, we provide estimations of the volume error due to the pixel size. In these simulations, we added random noise of -0.5 to 0.5 pixels to the segmented masks after straightening and prior to volume inference to take into account that worm widths are rounded to integer pixel values. This analysis shows that error due to the pixel size is small (maximum 0.6% for pharynx and 0.3% for body volume) and scales according to a power law with the volume.

We do not include this analysis in the revised manuscript, as the pixel size is only one source of noise and is included in our new measurements of the net instrument noise (see answer to point 1b below).

b. What is the net instrument noise due to pixel size, autofocusing, segmentation, and computationally straightening worms? What is their net contribution to the CV?

We now add additional experiments and analyses shown in Supplemental Figures 1c-f, 2, and 3c-d. that address this important point.

Using three independent approaches (see details as provided in the revised methods below), we estimate the net instrument noise for the body volume to be between 0.7% and 1.2%, depending on the larval stage. The net instrument noise for the pharynx volume is 4.8% (hatch), 1.9% (M1), 1.6% (M2), 1.6% (M3), and 1.3% (M4). Instrument noise for length is very similar to that of the volume.

This new analysis validates our speculation from our initial submission that instrument noise for pharyngeal volumes at hatch is higher than at other stages, which allows us to make more clear statements regarding this point in the revised text. Importantly, measurement noise does not impact our conclusions, and has a near negligible contribution to the observed heterogeneity for developmental stages from M1 onwards.

Specifically, we validate the following main conclusions:

(i) the biological heterogeneity of the pharynx is smaller than that of the body, also when subtracting noise from non-biological sources.

We correct the observed CV for a contribution instrument noise shown in the revised Figure 1c and in the new Supplemental Figures 1c-f. We can approximate the biological heterogeneity CV_{biol} as the square root of $(CV_{\text{observed}}^2 - CV_{\text{technical}}^2)$ (since the variance of the sum of two normal distributions is the sum of the variances of the individual distributions). Figure 1c shows that the CV of pharynx volumes is higher than the CV of body volumes, also after correcting for technical error. Supplemental Figures 1c-f now show that the impact of the technical error is very small, except for volumes at hatch:

(ii) the pharynx follows adder-like growth dynamics.

In the new Supplemental Figure 3c-d, we show that the technical noise does not explain the observed adder-like behavior and that the pharynx is indeed distinct from a pure folder. Specifically, we made the following two new analyses:

a) Simulations of folder model with technical noise. We ran simulations of a folder model for the pharynx, considering the effect of the technical noise. In the simulations, adding technical noise reduces the slope between ΔV and V_1 for a folder to be slightly less than 1. However, for none out of 10'000 iterations of the simulation, did the folder simulations result in slopes as small as what we

experimentally observed. We conclude that the pharynx has stronger size control than a folder and that the adder-like behavior is not due to instrument noise.

b) The apparent reduction in the correlation and slope between two variables x and y due to measurement noise is called attenuation bias and can be corrected for if the measurement noise in y is known and independent of x . In the new Supplemental Figure 4c, we show that the corrected slopes remain smaller than what is expected for a folder model and are close to an adder model.

Note that, in Supplemental Figure S4, we plot the relation between V_1 and V_2 (not between V_1 and ΔV like in the main figure). We do this in order to meet the requirement of independence of the variables for computing the attenuation bias. In this case the expected slope for an adder model is non-zero and indicated in blue in the graphs. The precise slope expected for an adder depends on the mean volume fold change undergone in a larval stage as follows: slope = $1/(\text{fold change})$. [Details regarding this relation were discussed in our previous work (Stojanovski et al., 2022)¹⁴].

We believe, these additional analyses of technical noise strengthen our manuscript and the conclusions drawn.

Detailed explanations of these analyses are provided in the revised methods (ln. 445 ff) and in Supplemental Figure 2:

Estimation of net instrument noise

Three independent approaches were taken to estimate the net instrument noise of pharynx and body volume and length. First, standard error of the regression to 10 neighboring point used to estimate volumes at hatch and molts revealed a precision of measurement for hatch to M4 for the pharynx of 4.8%, 1.9%, 1.6%, 1.6%, 1.3% band for the body of 1.2%, 0.96%, 0.82%, 0.70%, 0.87%. Second, volume trajectories were determined for 60 individuals at 5 minutes time resolution and split into two complementary sets, of which each had 10 minutes time resolution. The mean difference between the two measurement was close to the standard error of the linear regression. Third, an experiment was conducted, in which for 14 animals were imaged 20 times every 30 minutes. For each of the 20 images a separate autofocus was performed. The measurement error was determined as the median CV among repeated measurements at each time point.

Determination of trendlines in of ΔV vs. V_1 and comparison to simulations with technical noise

Trendlines in scatter plots were computed using robust linear regression using the `robustfit()` method of Matlab (v2021b) and default parameters.

To determine the impact of instrument noise, the CV of the biological heterogeneity was estimated by $\text{var}(\text{biological}) = \text{var}(\text{total}) - \text{var}(\text{technical})$. Starting volumes $V_{1, \text{biol}}$ were then drawn from a normal distribution with standard deviation corresponding to the biological heterogeneity and sample size corresponding to the number of individuals measured experimentally for the respective larval stage. A folder was simulated by multiplying with a fold-change FC_{biol} drawn from a normal distribution with standard deviation corresponding to the observed heterogeneity to yield a volume distribution $V_{2, \text{biol}}$. Subsequently, technical error was added to $V_{1, \text{biol}}$ and $V_{2, \text{biol}}$, drawn from a normal distribution with standard deviation of the technical noise determined for the respective larval stage, yielding simulated $V_{1, \text{biol} + \text{tech}}$ and $V_{2, \text{biol} + \text{tech}}$, from which $\Delta V_{\text{biol} + \text{tech}} = V_{2, \text{biol} + \text{tech}} - V_{1, \text{biol} + \text{tech}}$ was computed. $V_{1, \text{biol} + \text{tech}}$ and $\Delta V_{\text{biol} + \text{tech}}$ were normalized to their respective means, equivalent to the treatment of experimental data in Figure 3, and the slope of this relation was determined by linear regression. This simulation was conducted 10'000 times to compare simulations to the experimentally observed relation between with ΔV and V_1 .

Correction of regression slopes for attenuation bias

The slope of a linear regression between two observed variables $V_{1, \text{observed}}$ and $V_{2, \text{observed}}$ is biased towards zero due to measurement error in V_1 . To correct for this bias (called least squares attenuation bias), we computed the reliability ratio $\lambda = \frac{\text{var}(V_{1, \text{biol}})}{\text{var}(V_{1, \text{biol}}) + \text{var}(V_{1, \text{tech}})}$, where $\text{var}(V_{1, \text{biol}})$ is the variance of V_1 , corrected for the contribution of technical error: $\text{var}(V_{1, \text{biol}}) = \text{var}(V_{1, \text{observed}}) - \text{var}(V_{1, \text{tech}})$. $\text{var}(V_{1, \text{tech}})$ is the experimentally determined measurement error. The measured slopes between V_1 and V_2 were for then corrected for the attenuation bias by: $\text{slope}_{\text{corrected}} = \text{slope}_{\text{measured}} / \lambda$.

a Regression to time series

b Comparison two time series shifted by 5 minutes

c 20-times repeated sampling per time point

d

c. How are the errors propagated in the estimation of volume and does that account for the approximately 10-fold difference in the observed CV of pharynx length vs volume in Fig 1d vs Supplemental Fig 1c? Is there validation by comparing the estimates from taking multiple optical sections?

We answer these questions separately below.

Length vs. volume: The reviewer may have misread the CVs shown in the figures. Fig. 1d and Supplemental Fig. 1c show that, at all larval stages, the difference in CV between pharynx volume and length is 2-fold or less. We now show this data more clearly in the new Supplemental Figure 1c-f (see above). We note that the technical error of our length measurements is very similar to that of our volume measurements. However, since the biological heterogeneity of lengths is smaller than that of volumes, the technical error has a larger impact for length than for volume.

The volume is inferred by computing the volume of cylindrical slices of 1 pixel width. The length is determined as the total number of these slices, i.e. the length of the midline of the segmented worm. Technical variation in detecting the end points of the worm, has a larger impact on the length than on the volume since the thin tail of the worm adds very little to the volume. On the other hand, volume measurements are affected by noise in width measurements. Together, this may explain why the net instrument noise is very similar for length and volume.

Optical sections: Estimating volumes of shallow objects from optical sectioning is inherently challenging due to the poor axial resolution of fluorescence microscopy. Therefore, the rotational symmetry-based approach that we use, is broadly applied and accepted in the field for *C. elegans*^{23,24}, as well as other rotationally symmetric systems (e.g., bacteria²⁵, fission yeast²⁶), which we believe is a superior method to volumetric measurements by optical sectioning.

Below we provide validation of body volume measurements for L4 stage animals, where the limitation of axial resolution is least pronounced. We find good correlation between the planar measurements and optical sectioning. The figure shows volumes estimated from optical sectioning plotted against volumes estimated from planar measurements. Each circle is a different time point of a micro chamber experiment imaged every 10 minutes. Colors (yellow, blue, red) correspond to different individuals. This experiment was conducted on a spinning-disc confocal microscope (instead of a wide-field microscope) and at double the magnification of what we used in the manuscript (20x, 0.75 NA instead of 10x, 0.45 NA).

Since the measurements from optical sectioning are likely less precise than those from planar measurements, we feel these data would not significantly add to the manuscript. More importantly, our conclusions using volume estimates and length estimates are highly consistent, suggesting that we are not misled by technical aspects of size estimation.

While volume estimates are useful, the same points can be made with length data in Fig 1 with fewer caveats. Standardising on length would improve consistency across the entire manuscript.

We believe that volume is the appropriate measure for the first part of the manuscript (Figures 1 and 2) for the following reasons: In these figures, we compare the observed heterogeneity in volume to a null model of exponential volume growth. We feel it would not be justified to infer a growth model based on length alone, as the heterogeneity would also be impacted by fluctuations in the width-to-length aspect ratio, e.g., due to changes in worm posture. Moreover, as we show in the new supplemental Figures 2c and e, estimations of heterogeneities in length are more sensitive to technical noise. This is because the biological variability in length is smaller while the technical noise is similar for length and for volume.

The second section of the paper does not consider individual animals, but instead how the population mean varies across different conditions. This allows us to average over many individuals, such that individual-to-individual differences in the width-to-length aspect ratio do not impact our conclusions. Importantly, however, our conclusions also hold for volume-based analysis as shown in the supplemental figures.

2. Role of yap-1

There are some caveats to the analysis of yap-1 and potentially other models for yap-1's role.

a. The role of yap-1 is depicted as a major point in the manuscript based on the prominence in the title and abstract. However, the evidence for the role of yap-1 is solely based on RNAi, which may not be specific. It is important to validate this result with an outcrossed yap-1 mutant. Alternatively, the authors can be explicit about the caveats and reduce the prominence of yap-1 in the manuscript.

We now include additional experiments using a previously characterized and extensively outcrossed allele *yap-1(tm1416)* that deletes the WW domain of YAP-1. This mutant allele recapitulates the sensitivity of *yap-1(RNAi)* regarding deviations in pharynx size upon epidermal AID of RAGA-1. The mutant also shows good agreement with the RNAi for other phenotypes: growth rate, developmental speed, and pharyngeal length at L4 to adult transition.

Overall, the phenotype of the *yap-1(tm1416)* allele is slightly weaker than that of the RNAi. Most strikingly, the mutant allele does not cause larval arrest when combined with epidermal AID of RAGA-1. These data suggest that *tm1416* is not a complete null allele. Alternatively, larval arrest could be due to unspecific RNAi. We discuss these two possibilities in the revised manuscript.

The *yap-1(tm1416)* mutation enhances larval arrest of animals treated with *yap-1* RNAi (and epidermal RAGA-1 AID). This shows that the *yap-1(tm1416)* can cause larval arrest, albeit this function is only apparent when combined with *yap-1(RNAi)*. These data make us favor the interpretation that *tm1416* is not a complete null allele, rather than arrest being due to unspecific RNAi. Final evaluation of this point will require generation and detailed characterization of a full deletion allele by CRISPR/Cas 9 in future work.

Together, these results strengthen our evidence that *yap-1* is important for maintaining pharynx-to-body proportions as stated in the revised title and abstract.

b. To justify the title that yap-1 mediates ultra-sensitive coupling would require additional experiments to show that the Hill co-efficient is reduced when yap-1 activity is disrupted. Alternatively, this point can be addressed by revising the title to downplay this specific claim, which would in no way reduce the importance and interest of this work.

We agree with the reviewer and have adjusted the title accordingly, which now reads:

Maintenance of appropriate size scaling of the *C. elegans* pharynx by YAP-1

c. yap-1 signalling is presented as a general mechanism for robustness of proportionate growth. Is it not just as likely that yap-1 specific to the raga-1 pathway as there is literature relating mTOR signalling to Hippo/YAP (e.g., Csibi & Blenis, Nature Cell Biology 2012)? These alternative models should be mentioned in the discussion.

This is indeed a possibility, which we now discuss in ln. 380 of the revised manuscript:

In future research, it will be important to address if YAP-1 globally senses growth or size imbalances, or if its response is mediated by direct crosstalk between the mTOR and the Hippo pathway²⁷.

The reference cited is the primary article discussed by in the News & Views article of Csibi & Blenis.

Minor issues:

Does raga-1 affect RNAi efficacy? The authors suggest that yap-1 only comes into play when there is raga-1 perturbation. But there is no evidence provided to rule out the possibility that the RNAi is more severe under raga-1 loss. The point that other RNAi treatments do not synergize with raga-1 does not address this issue since those RNAi treatments may be true negatives. This caveat should be stated in the results section.

We now confirm the RNAi phenotypes using the genetic allele *yap-1(tm1416)* instead of RNAi (Figure 5). Since we do not use RNAi in these experiments, we can exclude that the stronger phenotypes of epidermal RAGA-1 AID strains is due to potential differences in RNAi efficiency.

Introducing sizing mechanisms (e.g., adder, folder) in greater detail would provide context to make it easier for readers to understand and appreciate many aspects of this work.

Adders and sizers are now introduced in greater detail in the second paragraph of the introduction (ln. 32 ff.):

At the scale of individual cells, size homeostasis has been extensively studied by time-lapse microscopy of yeasts, bacteria, and mammalian cells. In cells, stochastic size fluctuations are corrected within a few cell divisions. For many cell types, larger cells on average undergo a smaller volume fold change per cell cycle than smaller cells. Thereby, cells that deviate from the norm return to a stable reference point. Depending on how fast this reference point is reached, cells are called to follow adder or sizer mechanisms^{25,28–31}. A sizer refers to cell types that, on average, return to the appropriate size within one cell cycle such that their size at division is independent of their size at birth. An adder refers to cells that, on average, grow by a constant absolute volume, independent of their size at birth. Unlike sizers, adders take multiple cell cycles to return to a reference point.

Fig 2 could benefit from a color scale bar for a-b and c.

A color scale bar has been added.

Fig 3e – the sizes of the circles are obscured by the error bars. A larger panel could address this visualization issue.

We improved this visualization by showing the error bars in black and without cap. Instead of circle size, we use different marker types to indicate the IAA concentration.

Typo in Fig 2 legend: "...produces and adder-like..."

DONE.

Typo in Supplemental Fig 2c-d y-axis label: "voulme"

DONE.

References

1. Vea, I. M. & Shingleton, A. W. Network-regulated organ allometry: The developmental regulation of morphological scaling. *WIREs Developmental Biology* **10**, e391 (2021).

2. West, G. B., Brown, J. H. & Enquist, B. J. A General Model for the Origin of Allometric Scaling Laws in Biology. *Science* **276**, 122–126 (1997).
3. Almuedo-Castillo, M. *et al.* Scale-invariant patterning by size-dependent inhibition of Nodal signalling. *Nat Cell Biol* **20**, 1032–1042 (2018).
4. Ben-Zvi, D. & Barkai, N. Scaling of morphogen gradients by an expansion-repression integral feedback control. *Proceedings of the National Academy of Sciences* **107**, 6924–6929 (2010).
5. Mateus, R. *et al.* BMP Signaling Gradient Scaling in the Zebrafish Pectoral Fin. *Cell Reports* **30**, 4292–4302.e7 (2020).
6. Hamaratoglu, F., de Lachapelle, A. M., Pyrowolakis, G., Bergmann, S. & Affolter, M. Dpp signaling activity requires Pentagone to scale with tissue size in the growing *Drosophila* wing imaginal disc. *PLoS Biol* **9**, e1001182 (2011).
7. Boulan, L. & Léopold, P. What determines organ size during development and regeneration? *Development* **148**, (2021).
8. Vollmer, J., Casares, F. & Iber, D. Growth and size control during development. *Open Biology* **7**, (2017).
9. Averbukh, I., Ben-Zvi, D., Mishra, S. & Barkai, N. Scaling morphogen gradients during tissue growth by a cell division rule. *Development* **141**, 2150–2156 (2014).
10. Colombani, J., Andersen, D. S. & Léopold, P. Secreted Peptide Dilp8 Coordinates *Drosophila* Tissue Growth with Developmental Timing. *Science* **336**, 582–585 (2012).
11. Garelli, A., Gontijo, A. M., Miguela, V., Caparros, E. & Dominguez, M. Imaginal Discs Secrete Insulin-Like Peptide 8 to Mediate Plasticity of Growth and Maturation. *Science* **336**, 579–582 (2012).
12. Roselló-Díez, A., Madisen, L., Bastide, S., Zeng, H. & Joyner, A. L. Cell-nonautonomous local and systemic responses to cell arrest enable long-bone catch-up growth in developing mice. *PLOS Biology* **16**, e2005086 (2018).
13. Baron, J. *et al.* Catch-up growth after glucocorticoid excess: a mechanism intrinsic to the growth plate. *Endocrinology* **135**, 1367–1371 (1994).
14. Stojanovski, K., Großhans, H. & Towbin, B. D. Coupling of growth rate and developmental tempo reduces body size heterogeneity in *C. elegans*. *Nature Communications* **13**, 3132 (2022).
15. Zhang, L., Ward, J. D., Cheng, Z. & Dernburg, A. F. The auxin-inducible degradation (AID) system enables versatile conditional protein depletion in *C. elegans*. *Development* **142**, 4374 (2015).
16. Sancak, Y. *et al.* Ragulator-Rag Complex Targets mTORC1 to the Lysosomal Surface and Is Necessary for Its Activation by Amino Acids. *Cell* **141**, 290–303 (2010).
17. Binda, M. *et al.* The Vam6 GEF controls TORC1 by activating the EGO complex. *Mol Cell* **35**, 563–573 (2009).
18. Smith, H. J. *et al.* Neuronal mTORC1 inhibition promotes longevity without suppressing anabolic growth and reproduction in *C. elegans*. 2021.08.12.456148 (2021) doi:10.1101/2021.08.12.456148.
19. Lee, H., Kang, J., Ahn, S. & Lee, J. The Hippo Pathway Is Essential for Maintenance of Apicobasal Polarity in the Growing Intestine of *Caenorhabditis elegans*. *Genetics* **213**, 501–515 (2019).
20. Lee, H., Kang, J. & Lee, J. Involvement of YAP-1, the Homolog of Yes-Associated Protein, in the Wnt-Mediated Neuronal Polarization in *Caenorhabditis elegans*. *G3 Genes/Genomes/Genetics* **8**, 2595–2602 (2018).
21. Iwasa, H. *et al.* Yes-associated protein homolog, YAP-1, is involved in the thermotolerance and aging in the nematode *Caenorhabditis elegans*. *Experimental Cell Research* **319**, 931–945 (2013).
22. Teuscher, A. C. *et al.* Mechanotransduction coordinates extracellular matrix protein homeostasis promoting longevity in *C. elegans*. *bioRxiv* 2022.08.30.505802 (2022) doi:10.1101/2022.08.30.505802.
23. Uppaluri, S. & Brangwynne, C. P. A size threshold governs *Caenorhabditis elegans* developmental progression. *Proceedings of the Royal Society B: Biological Sciences* **282**, (2015).
24. Moore, B. T., Jordan, J. M. & Baugh, L. R. WormSizer: High-throughput Analysis of Nematode Size and Shape. *PLOS ONE* **8**, e57142 (2013).

25. Taheri-Araghi, S. *et al.* Cell-Size Control and Homeostasis in Bacteria. *Current Biology* **25**, 385–391 (2015).
26. Miller, K. E., Vargas-Garcia, C., Singh, A. & Moseley, J. B. The fission yeast cell size control system integrates pathways measuring cell surface area, volume, and time. *Current Biology* **33**, 3312–3324.e7 (2023).
27. Tumaneng, K. *et al.* YAP mediates crosstalk between the Hippo and PI(3)K–TOR pathways by suppressing PTEN via miR-29. *Nat Cell Biol* **14**, 1322–1329 (2012).
28. Campos, M. *et al.* A Constant Size Extension Drives Bacterial Cell Size Homeostasis. *Cell* **159**, 1433–1446 (2014).
29. Soifer, I., Robert, L. & Amir, A. Single-Cell Analysis of Growth in Budding Yeast and Bacteria Reveals a Common Size Regulation Strategy. *Current Biology* **26**, 356–361 (2016).
30. Xie, S. & Skotheim, J. M. A G1 Sizer Coordinates Growth and Division in the Mouse Epidermis. *Current Biology* **30**, 916–924.e2 (2020).
31. Schmoller, K. M. & Skotheim, J. M. The Biosynthetic Basis of Cell Size Control. *Trends in Cell Biology* **25**, 793–802.

REVIEWERS' COMMENTS

Reviewer #1 (Remarks to the Author):

The authors have made the changes requested (by both reviewers). The clarifications with regard to the model, the change of title, and noise quantification, along with the improved visualisations, and additional explanations have significantly improved the manuscript.

I recommend publication in nature communications.

Reviewer #2 (Remarks to the Author):

The authors have fully addressed all my comments. I thank them for their extensive experiments and analyses that have strengthened the manuscript considerably.

One loose end: For consistency, it would be great to add labels for wild type and yap-1(tm1416) in Figure 5d.

Point by point answer to reviewer's requests

Reviewer #1 (Remarks to the Author):

The authors have made the changes requested (by both reviewers). The clarifications with regard to the model, the change of title, and noise quantification, along with the improved visualisations, and additional explanations have significantly improved the manuscript.

I recommend publication in nature communications.

We thank the reviewer for these supportive words about our work and for the valuable feedback on our initial submission

Reviewer #2 (Remarks to the Author):

The authors have fully addressed all my comments. I thank them for their extensive experiments and analyses that have strengthened the manuscript considerably.

One loose end: For consistency, it would be great to add labels for wild type and yap-1(tm1416) in Figure 5d.

We thank the reviewer for these supportive words about our work and for the valuable feedback on our initial submission. We have now added the relevant label to Fig. 5d.